# Impact hotspots of reduced nutrient discharge shift across the globe with population and dietary changes

Xu Wang [1,2], Glen Daigger[3], Wim de Vries[4,5], Carolien Kroeze[6], Min Yang[7,8], Nan-Qi Ren[9], Junxin Liu[1,8] & David Butler [2]

Reducing nutrient discharge from wastewater is essential to mitigating aquatic eutrophication; however, energy- and chemicals-intensive nutrient removal processes, accompanied with the emissions of airborne contaminants, can create other, unexpected, environmental consequences. Implementing mitigation strategies requires a complete understanding of the effects of nutrient control practices, given spatial and temporal variations. Here we simulate the environmental impacts of reducing nutrient discharge from domestic wastewater in 173 countries during 1990–2050. We find that improvements in wastewater infrastructure achieve a large-scale decline in nutrient input to surface waters, but this is causing detrimental effects on the atmosphere and the broader environment. Population size and dietary protein intake have the most significant effects over all the impacts arising from reduction of wastewater nutrients. Wastewater-related impact hotspots are also shifting from Asia to Africa, suggesting a need for interventions in such countries, mostly with growing populations, rising dietary intake, rapid urbanisation, and inadequate sanitation.

---

[1] Research Center for Eco-Environmental Sciences, Chinese Academy of Sciences, 100085 Beijing, China. [2] Centre for Water Systems, College of Engineering, Mathematics and Physical Sciences, University of Exeter, Exeter EX4 4QF, United Kingdom. [3] Department of Civil and Environmental Engineering, University of Michigan, Ann Arbor, MI 48109, United States. [4] Wageningen Environmental Research, Wageningen University & Research, 6700 AA Wageningen, Netherlands. [5] Environmental Systems Analysis Group, Wageningen University & Research, 6700 AA Wageningen, Netherlands. [6] Water Systems and Global Change Group, Wageningen University & Research, 6700 AA Wageningen, Netherlands. [7] State Key Joint Laboratory of Environment Simulation and Pollution Control, Research Center for Eco-Environmental Sciences, Chinese Academy of Sciences, 100085 Beijing, China. [8] University of Chinese Academy of Sciences, 100049 Beijing, China. [9] State Key Laboratory of Urban Water Resource and Environment, School of Environment, Harbin Institute of Technology, 150090 Harbin, China. Correspondence and requests for materials should be addressed to X.W. (email: xuwang@rcees.ac.cn)

Most human use of water produces wastewater. Substantial increases in the global population and economic activity have increased wastewater production, which exceeded 300 billion tonnes per year since 2010[1]; which is equivalent to approximately 8% of all freshwater withdrawals[2]. The continuing growth and spread of cities, combined with insufficient and inadequate sanitation facilities and poor management, implies that a vast proportion of wastewater could enter and affect aquatic ecosystems[3]. Over the past half-century, domestic wastewater (in addition to food production) has been a key contributor to nutrient increases in the aquatic environment[4]. Global nitrogen (N) and phosphorus (P) flows currently exceed planetary thresholds[5], whereas the quantities of N or P passing into domestic wastewater from human metabolism have become key components in global N and P cycles[6,7]. For instance, N in human excreta represents roughly 15–20% of the total anthropogenic production of reactive N per year[1].

Significant investments have been made in the wastewater industry to meet the regulations restricting point-source nutrient discharge[8]. In most developing countries, these regulations are consensus-based, emissions standards set at a generic level[9]. In parts of the developed world, such as Australia, Europe, and North America, where eutrophication is of particular concern[10], more-stringent standards have been implemented to regulate nutrient removal. Furthermore, wastewater management infrastructure, which, in the recent past, has provided effective point-source solutions to protect public health and the aquatic environment, can be retrofitted to enhance the treatment levels and to remove more nutrients.

However, less emphasis has been placed on the broader environmental responses to the growing number of nutrient discharge interventions. The use of fossil fuels and the resultant emissions of airborne contaminants (such as $CO_2$, $CH_4$, and $N_2O$) in the nutrient removal process, for example, can have other, seemingly unrelated, environmental impacts[11,12]. These patterns cause increasingly pressing challenges related to the nature and significance of the environmental impacts of such interactions[13]. This relates particularly to whether and how wastewater management affects the regulation of biogeochemical processes in the biosphere and ensures sustainability. Addressing such issues requires a systems-based approach coupled with various human and natural components that interact across technologies, space, and time[14]. Such an approach allows an understanding of the interconnectivity and complexity of the entire system.

We evaluated the environmental impacts of reducing point-source nutrient discharge and the use of domestic wastewater facilities capable of achieving such nutrient removal. In addition, we identified factors that could play a vital role in driving these impacts and we quantified the extent to which potential synergies and trade-offs could be achieved from a global systems perspective. We based our study on a transparent, integrated model that allowed us to vary the key demographic and socioeconomic driving factors of future wastewater management services, while considering both temporal and spatial variations. We traced and visualised shifts in environmental effects, unravelling the potential patterns that could occur from changing wastewater management practices. Sensitivity analyses showed how these responses would interact with potential changes in population, urbanisation, diet, and sanitation infrastructure, among other future socioeconomic condition changes by 2050. Finally, we combined our results with reported thresholds on planetary boundaries[15] to assess the overall global impact of wastewater management trajectories.

## Results

**Overview.** We modelled three successive time periods (1990–2050), each with a distinct wastewater management regime, to test the magnitude of the impacts in relation to the enhancement of nutrient removal through the development of wastewater infrastructure. The business-as-usual (BAU) regime, in particular, aligns the prevalence of different wastewater treatment levels (primary and secondary) with their respective estimates at regional level to simulate the actual nutrient removal situations for the period 1990–2010. The secondary-treatment-for-all (STA) regime assumes universal access to secondary wastewater treatment for the period 2011–2030. The universal-tertiary-treatment (UTT) regime represents a more-optimistic likelihood for future nutrient removal and assumes complete usage of tertiary treatment infrastructure across the globe for the period 2031–2050. The assumptions made for the regimes are presented in detail in the Methods section.

To ensure the representativeness of the analyses, we included various wastewater treatment technologies to attain the different standards for nutrient removal. Building upon the modelling methodology (Supplementary Fig. 1), we evaluated various environmental impacts of nutrient discharge and the implementation of wastewater facilities capable of achieving the target removal levels. The environmental impact categories include the N and P cycles, climate change (in terms of greenhouse gas emissions), stratospheric ozone depletion, atmospheric aerosol loading, chemical pollution, biodiversity loss, and freshwater use. The impacts attributed to the three regimes were evaluated in 173 countries from Africa, the Americas, Asia, Oceania, and Europe by integrating the national trajectories for varying leading parameters, including population, urbanisation, food- and water-(consumption) related factors, and sanitation development. All the values, parameters, and sources for the model settings are provided in the Methods and the Supplementary Information. A summary of the global ranges of the model outputs in 2010, 2030, and 2050 is presented in Supplementary Fig. 2.

**Changing nutrient discharges from domestic wastewater.** Firstly, we traced the increases in both N and P inputs to surface waters at a global scale within the distinct timeframe of each wastewater management regime (Supplementary Data 1). The overall global outcomes for the last year of each regime suggested that improvements in wastewater treatment services could have a positive effect on reducing N and P input to the aquatic environment (Fig. 1a). In 2010, the mid-range N and P flows under the BAU regime were nearly 8.7 teragram (Tg = $10^{12}$ g) yr$^{-1}$ and 2.1 Tg yr$^{-1}$, respectively. After achieving global prevalence of secondary wastewater treatment in the STA regime, the overall N and P flows declined slightly to 8.6 Tg yr$^{-1}$ and 1.9 Tg yr$^{-1}$, respectively, in 2030. This implies that the extensive application of secondary treatment facilities are insufficient to mitigate nutrient discharge from domestic wastewater adequately because of the ever-rising global population and other negative factors. By 2050, the N and P flows decreased sharply to 3.7 Tg year$^{-1}$ and 0.8 Tg year$^{-1}$, respectively, indicating that nutrient reduction occurred under the UTT regime, with the universal employment of tertiary facilities, despite the global population projection by 2050 being 38 and 16% larger than that in 2010 and 2030 (ref. [16]), respectively.

Furthermore, the results revealed spatial heterogeneity between and within continents (Fig. 1b). As similar patterns were seen in both N and P flows, we focused on further exploration of the N flows (for full model outputs, see Supplementary Data 1). We calculated the per capita N discharge to isolate the influence of the population base (Fig. 1c). Notably, high per capita N

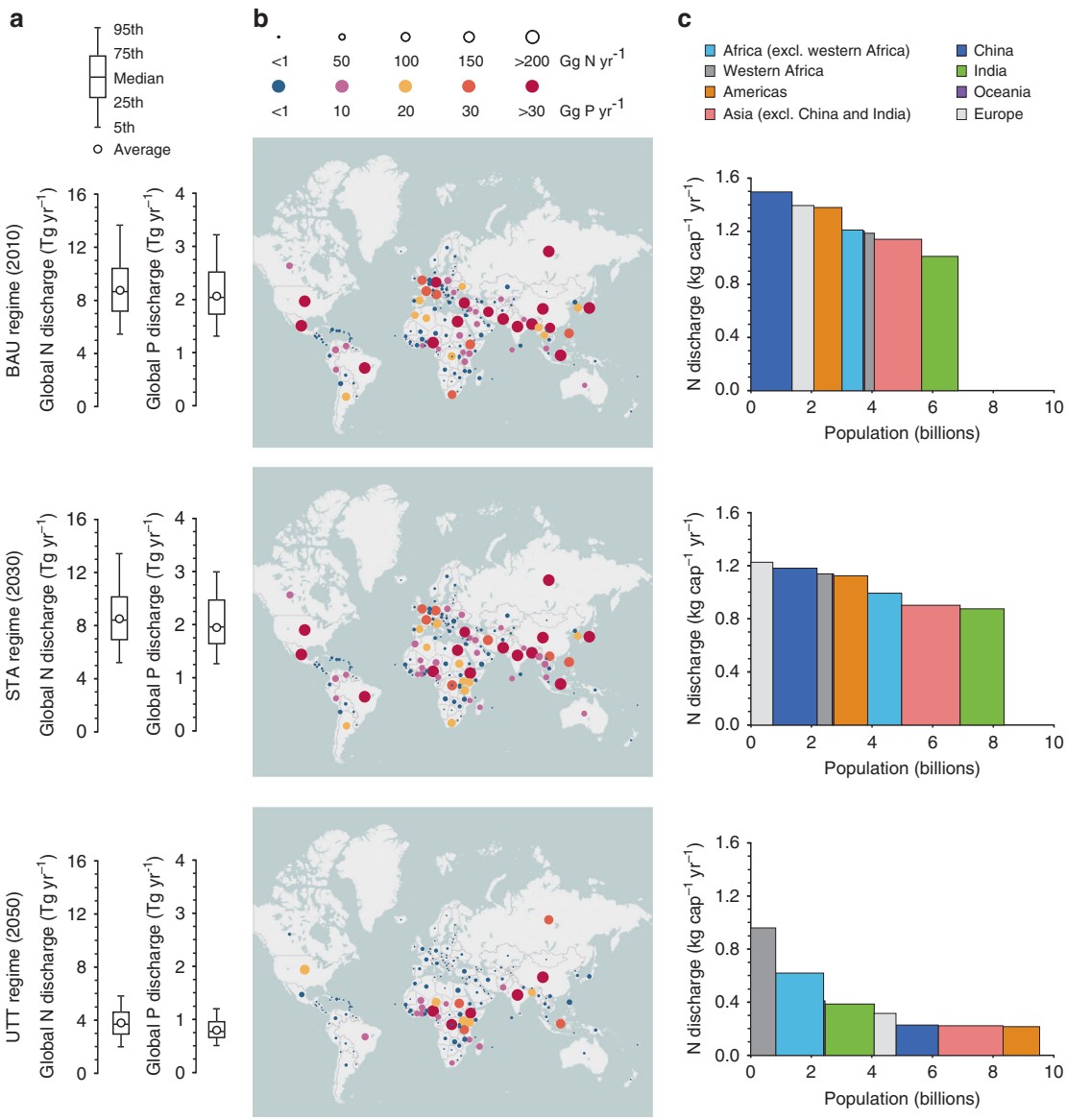

**Fig. 1** Spatial patterns of nitrogen and phosphorus flows from domestic wastewater management in the three regimes. **a** Box-whisker plots show the range of global total N and P flows. **b** Circles in the maps indicate 173 countries. Circle sizes indicate N emissions (Gg N yr$^{-1}$), and their colours indicate P emissions (Gg P yr$^{-1}$). **c** Column heights represent per capita N discharge, and column areas denote total N emissions from their respective regions. Mid-range outputs are used to generate the charts of **b** and **c**

discharge is correlated commonly with intensive dietary protein intake, low usage of sanitation services, inadequate and insufficient nutrient removal facilities, and combinations of these factors (Supplementary Fig. 3).

With increasing urbanisation and growing wealth, people generally increase their nutrient intake and, particularly, the portion of protein-intensive food in their diet[17]. In 2010, the estimated global average protein consumption was ~68 g per capita per day, whereas in the wealthier economies in Europe and North America, protein consumption was higher at a mid-range value of ~104 g protein cap$^{-1}$ d$^{-1}$ (ref. [18]). Consequently, dietary protein intake could be a major driving factor in the high per capita N discharges observed in Europe and the Americas (Fig. 1c), as improved sanitation and wastewater facilities to implement nutrient removal have been practiced relatively well in these regions. Nevertheless, these developed regions were responsible for only small portions of the global N input from domestic wastewater (for instance, only 15% from the Americas,

12% from Europe, and less than 1% from Oceania in 2010; Supplementary Table 1), after scaling up their per capita emission factors with the low population bases.

However, developing and emerging economies present more-complex situations. For instance, although the global trend is towards overconsumption of calories and protein-rich foods[17], the average daily per capita protein intake in a significant number of countries and regions was lower than the global average consumption in 2010, with most of these countries and regions being located in Africa and Asia[18]. However, we found that these low-protein consumption regions usually collocated insufficient sanitation and wastewater management (Supplementary Fig. 3). This situation could explain why the per capita N discharges in Africa and Asia are comparable with those in the developed economies in 2010 (Fig. 1c). Although we made optimistic assumptions for the future possibility of achieving worldwide tertiary wastewater treatment by 2050, the per capita N emissions remained higher across the less-developed world, particularly in

African regions, such as western Africa. Along with rising populations, the continent of Africa alone will contribute half of the global total N input by 2050 (1.8 Tg N yr$^{-1}$, Supplementary Table 1). In 2010, Asian countries together accounted for most of the global N input arising from the same wastewater treatment process (5.1 Tg N yr$^{-1}$, Supplementary Table 1), partly owing to their large population bases. However, the per capita N emissions from domestic wastewater in emerging economies such as China and India are expected to decrease significantly along with substantial improvements in sanitation services and wastewater treatment practices from 2010 to 2050 (Fig. 1c). However, compared with China during this period, the rate of decease in N input was lower in India (Supplementary Table 1) because of the lower standard of sanitation, while India would have a larger population than China by 2050.

Most national, regional, or global projections of anthropogenic nutrient discharge rely heavily on assumptions and calculations rather than observations or experiments[19–21]. Consequently, there are significant disparities in these reported estimates; therefore, using one estimate to calibrate or verify another estimate is unrealistic. Accordingly, our model to estimate nutrient inputs from domestic wastewater was not calibrated against the figures reported in the literature. Nonetheless, we did compare the nutrient discharge estimates from this study with those from the published research. Our overall global estimates of nutrient discharge from domestic wastewater were noticeably lower than were the projections in a study by van Drecht et al.[19] (2.5–5.7 vs. 12.0–15.5 Tg N yr$^{-1}$ and 0.5–1.2 vs. 2.4–3.1 Tg P yr$^{-1}$ by 2050). The algorithm we used to determine nutrient discharge levels was comparable with this previous study to some extent because the calculation principle was similar. The calculation principle was, namely, estimating nutrient effluent quantities based on per capita dietary protein intake and, subsequently, scaling according to general access to sanitation and wastewater services. The differences could be explained mainly by the different assumptions used in the two estimates for the model settings and value sets. Van Drecht et al.[19] assumed that only part of the wastewater flows connected to public sewerage systems would be treated eventually by centralised facilities, and the rest of the anthropogenic nutrients would be discharged into the environment without treatment or reuse. In contrast, our model assumed that all the nutrients from human excreta would be disposed of and recycled further, depending on their treatment in centralised or decentralised facilities and the degree of treatment they received (described in detail in the Methods section). Therefore, it is not surprising that the work of van Drecht et al.[19] projected lower estimates of both N and P discharges. Moreover, van Drecht et al.[19] used relatively high values for dietary protein intake and estimated lower levels of nutrient reduction, which partly explains the larger nutrient discharge estimates obtained. However, we estimated a range of N output of 1.3–3.2 Tg N yr$^{-1}$ from domestic wastewater facilities in China in 2010, which overlaps with the range of estimates in other studies (0.8–1.8 Tg N yr$^{-1}$; ref. [20,21]). This shows that nutrient discharges from different systems and regions can be comparable only if standardised methods were adopted.

**Environmental impacts of improvements in wastewater services.** Figure 2 shows an overview of the normalised results of the broad environmental impacts of the three wastewater management regimes, including changes in influences across geographic regions and over time. As regards the nutrient cycles (Fig. 2a, b), the nutrient discharge hotspots arising from domestic wastewater were transferring from eastern and south-central Asia to western Africa throughout the time period, reaching a peak by 2050. This finding is consistent with the analyses of the N flows shown in Fig. 1. Further, we observed distinct boundaries between the colour zones of STA and UTT in the charts of both N and P cycles (Fig. 2a, b), indicating the significance of universal access to tertiary treatment in altering the nutrient discharge patterns of wastewater treatment systems.

In addition, Fig. 2 demonstrates that the impacts move gradually from affecting nutrient cycles to affecting climate change (in terms of greenhouse gas emissions), atmospheric aerosol loading, chemical pollution, freshwater use, ozone depletion, and biodiversity loss. The change pattern across the nutrient cycles and other broad impact categories shown in Fig. 2 are relevant particularly to Asia, the Americas, Europe, and Africa. Probably, although most countries in eastern and south-central Asia were releasing a decreasing amount of nutrients into the aquatic environment from domestic wastewater (the related areas turning into blue colours in Fig. 2), negative consequences still ensued, particularly in relation to climate change and atmospheric aerosol loading. This suggested that shifts in environmental impacts could occur along with the improved nutrient removal practices in these regions. In fact, the rates of increase in both sanitation and population factors in developed economies, such as Canada, the United States, and most European countries remained lower and more constant. Accordingly, it is not surprising that the changes in the impact categories could be observed only as their wastewater management regimes were transitioning from BAU to STA and UTT regimes (Fig. 2). Wastewater management in the rest of the less-developed world, particularly in Africa, require considerably more focus, as enhancements in sanitation and wastewater services produce marginal benefits in reducing burdens on nutrient cycles and the broad environment (Fig. 2). The full sets of the normalised values are presented in Supplementary Data 2.

**Key drivers in the transition of impact hotspots.** To isolate the multiple driving factors crucial to actuating the impacts of wastewater management services, we conducted sensitivity analyses for various demographic and socioeconomic drivers (the details are presented in the Methods section). We varied each input by ±20%, in turn, observing the effect on the environmental impact categories and various combinations thereof, using a relative sensitivity index (RSI). The key RSI results at a global scale are presented in Fig. 3, with a complete version with country-level values presented in Supplementary Data 3.

Figure 3 indicates that the results were extremely sensitive to various assumptions, especially those on population, per capita dietary protein intake, and the N content of dietary protein, sanitation usage, and nutrient removal efficiency. Furthermore, slight sensitivity was shown for assumptions regarding the prevalence of secondary treatment, urbanisation, and dietary structure factors, but lower sensitivity for assumptions on the parameters related to water withdrawal and consumption for municipal use. In the following sections, we explore why these factors were significant, how they contributed to the environmental impacts, in which countries or regions they could play the most crucial role, and how these implications might be useful to individuals, utilities managers, and policy-makers endeavouring to minimise the environmental impacts associated with point-source nutrient discharges, as well as measures to reduce such unwanted effects.

Population and urbanisation: According to United Nations (UN) estimates, the global population will reach 9.5 billion by 2050[16]. Such population growth will increase vastly the volumes of wastewater and the quantities of related contaminants. As suggested in Fig. 3, population size has the most significant effect

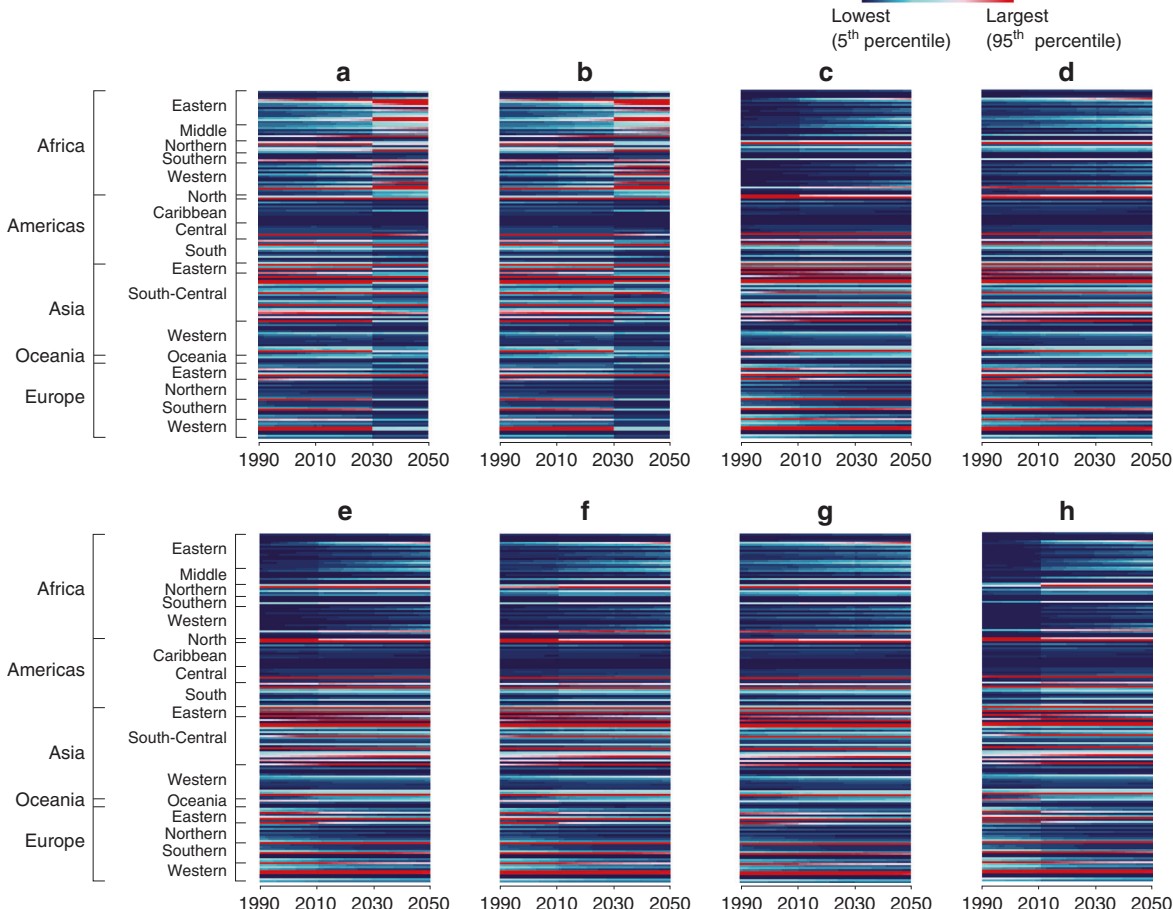

**Fig. 2** Temporally and spatially differentiated normalised environmental impact hotspots. **a** Nitrogen cycle; **b** Phosphorus cycle; **c** Climate change; **d** Stratospheric ozone depletion; **e** Atmospheric aerosol loading; **f** Chemical pollution; **g** Biodiversity loss; and **h** Freshwater use. To acquire these normalised results of the annual regional-scale trajectories of environmental impacts from wastewater treatment, the scores for each impact category, year, and country are divided by the global impact scores for the same year, providing a dimensionless ratio. The colour scale ranks the magnitude of the impacts (with magenta denoting a significant impact hotspot) and reveals the shifts in pressures (from navy-blue to magenta)

across all the environmental impact categories. Accordingly, it could be expected that wastewater management in countries and regions with large populations would create intensification of negative environmental impacts. For instance, the per capita N discharge in China under the UTT regime is predicted to be comparable with that in the Americas by 2050; however, as China has a larger population, the country alone could have a larger share than the Americas in global N flows from domestic wastewater (Fig. 1 and Supplementary Table 1). In addition, the spread of the different populations across the planet has an effect on the environment. Whereas developed countries have lower birth rates, developing countries have higher birth rates owing to poverty and less access to family planning and education. Consequently, it is estimated that by 2050, nearly 90% of the global population will live in less-developed countries;[16] although it may be worth noting that this percentage apparently did not take into consideration China's very fast development and its aspiration to be a member of the developed world by the middle of this century. These rapidly growing populations could create significant environmental pressure in the developing regions, as indicated by the higher impact figures, especially in western Africa (Figs. 1 and 2).

Furthermore, worldwide, populations are becoming more urbanised, with two-thirds of the global population projected to live in cities by 2050[16]. As sanitation in urban areas is often

superior to that in rural communities[22], it could be expected that urbanisation would contribute to a certain extent to reducing nutrient inputs from domestic wastewater to the environment. However, as shown in Fig. 3, other environmental impact factors show higher sensitivity to the assumptions about urbanisation than do the nutrient effects. This indicated that any benefit would be accompanied by shifts in impact, namely, climate change and atmospheric aerosol loading would become significant challenges because of increases in energy consumption, chemical usage, and airborne contaminant emissions arising from the improved sanitation and wastewater services in urban areas[12].

Daily protein intake and dietary structure: Rising incomes and increased awareness of the crucial role of protein in human health and well-being are stimulating the demand for high-protein foods[23]. As shown in Fig. 3 of this study, in addition to the population driving factor, total dietary protein intake had the most significant influence across all the environmental impacts. We also observed similar trends and magnitudes while analysing the effect of the N content of protein on the model outputs. However, protein typically contains ~13% of N by mass[24], which could not likely and easily be changed without future advancements in food technologies. In contrast, the percentage of P content in protein could vary significantly with various types of meat- and plant-based protein;[24] moreover, the wastewater-related impacts were more sensitive to the P content of plant-

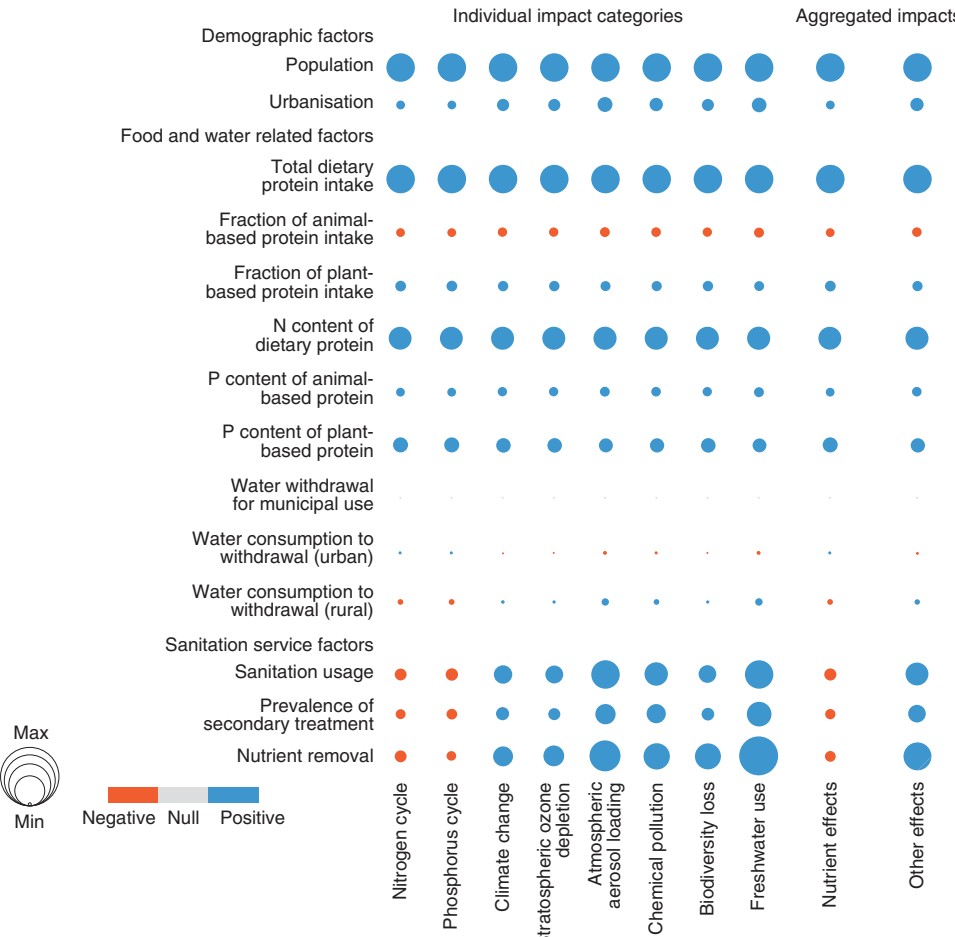

**Fig. 3** Sensitivity analysis for different driving factors. Circle size indicates RSI. The larger the circle size, the more sensitive the model output is to the input parameter. Circle colour indicates the relationship between model output and input (orange represents negative while blue represents positive). The sensitivity of aggregated impacts is determined as the average of the RSI of relative individual impacts

based protein (Fig. 3). Furthermore, we found that dietary changes could alter the environmental impacts resulting from wastewater service activities. When the total daily protein consumption was kept constant, increasing the fraction of meat-based protein was shown to reduce the negative environmental impacts of wastewater management services—quite the opposite to increasing the fraction of plant-based protein intake (Fig. 3). This result could be explained by assumptions that plant-based protein usually contained twice as much P as meat-based protein[24].

Sanitation and wastewater services: Reducing aquatic nutrient discharges relies mainly on decreasing the concentrations of nutrients released into the environment and capturing and treating waste streams[25]. The significance of improved sanitation infrastructure to controlling nutrient discharges and the spread of diseases in less-developed countries has been recognised by the international community in the coverage targets set by the UN Agenda 2030[26]. Continuous efforts have been made by the wastewater industry to satisfy the regulations limiting point-source nutrient discharge[9]. Figure 4 shows that sanitation usage, the prevalence of nutrient removal facilities (at least secondary treatment), and nutrient removal efficiency are the factors that stimulate the strengthening and reshuffling of the environmental impact hotspots in a relatively straightforward manner. Furthermore, we found that improved overall usage of sanitation infrastructure could be more efficient than increments in the prevalence of secondary treatment facilities and nutrient removal

efficiencies to reduce domestic wastewater burdens on the nutrient cycles. However, increasing the nutrient removal efficiencies of wastewater facilities could lead to an increase in unwanted effects on the broad environment. This finding highlighted a key negative environmental trade-off, namely that improving the quality of the receiving water (reducing nutrient inputs) could come at the expense of increased costs for service facility construction, increased electricity and chemicals consumption during facility operation, increased greenhouse gas emissions, and larger water footprints throughout these processes.

**The perspective of One Earth on wastewater nutrient control.** The depletion of the natural resources that sustain human well-being and prosperity, as well as the other anthropogenic impacts of global environmental change, are occurring at an unprecedented rate[27]. Research on such large-scale changes and their consequences for human well-being has led to the development of a set of planetary boundaries, representing the estimated thresholds for all anthropogenic activities, to guide the governance of the Earth system[15]. Over the past hundred years, domestic wastewater treatment methods have been improved and systems have been retrofitted to ensure the safety of aquatic ecosystems and to minimise the risks to human health[28]. When we applied the concept of planetary boundaries[15] as thresholds to benchmark our environmental impact results (Fig. 4a), we found that the N and P flows deriving from domestic wastewater management

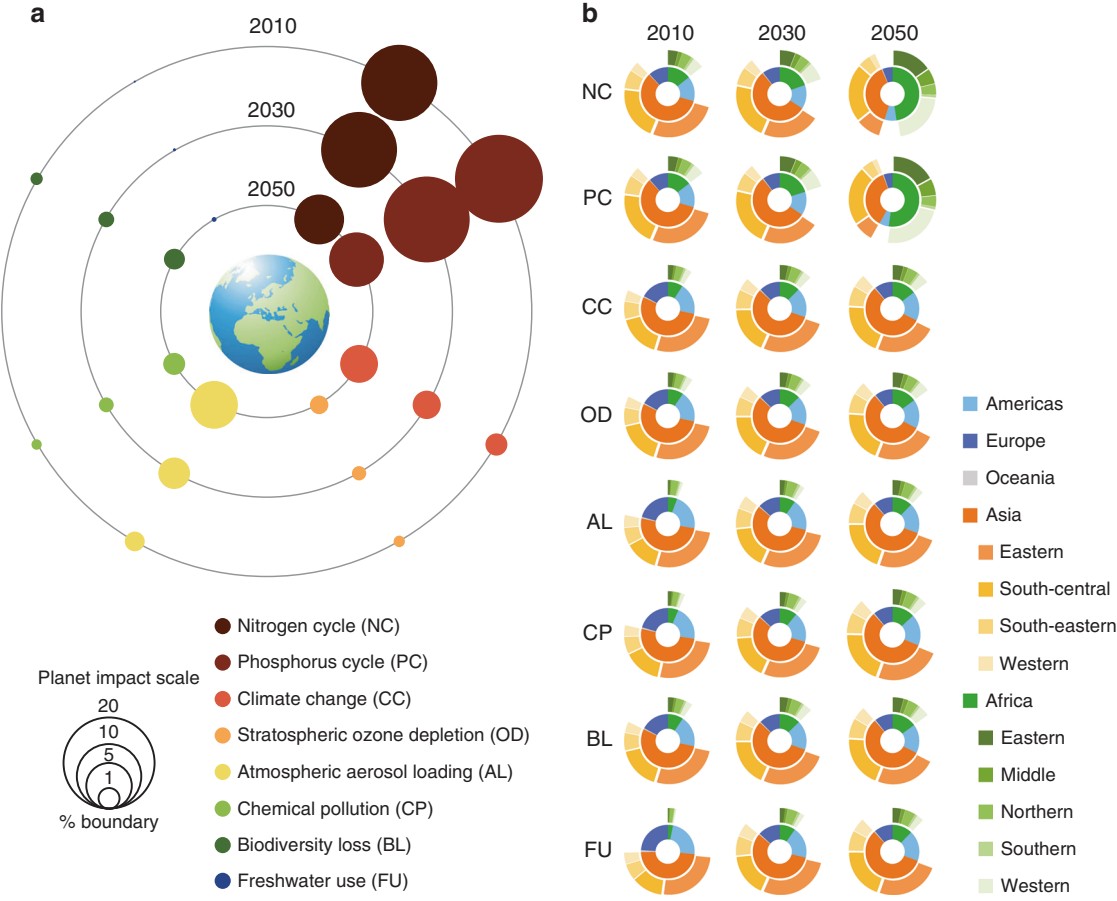

**Fig. 4** Implications of wastewater management trajectories in terms of global thresholds on planetary boundaries. **a** Circles represent the overall global effects attributed to wastewater services, and their size represents the planetary impact ratio. **b** Ring diagrams trace the contributions of the continents to the effects. The following reported thresholds for the planetary boundaries are used to normalise the effect scores into dimensionless ratios: 62 Tg N yr$^{-1}$ (NC); 11 Tg P yr$^{-1}$ (PC); $+1.0$ W m$^{-2}$ (CC); 14 DU yr$^{-1}$ (OD); 14.57 Tg PM$_{10}$-eq (AL); 4.36 million DALYs (CP); 280,729 species yr yr$^{-1}$ (BL); and 4000 km$^3$ yr$^{-1}$ (FU)

reached nearly 14 and 19% of their respective global planetary boundaries in 2010. In the current analysis, these portions decreased based on the optimistic assumption of the possible future achievement of global efficient nutrient removal facilities (UTT regime), cutting down the percentages to ~6 and 7% by 2050 (Fig. 4a). This implies that more-stringent point-source regulations and enhanced technological interventions on nutrient discharge from domestic wastewater do benefit the quality of aquatic systems, particularly in freshwater systems, where P pollution is recognised as the most significant challenge[29]. However, it should be noted that the extent of overall anthropogenic activity has already exceeded the planetary boundaries for both the N and P cycles[5]. Although enhanced intervention of nutrient discharge from domestic wastewater deals with only a small part of the anthropogenic impacts on nutrient boundaries (Fig. 4a), denitrification processes in wastewater treatment services could redirect dinitrogen gas into the atmosphere, offsetting some of what is currently extracted by humans. Furthermore, recycling N as fertiliser could function in the same way, reducing N fixation directly by decreasing the demand for industrially produced fertiliser.

Figure 4a shows the reshuffling of the environmental impact hotspots. Although the pressures on the nutrient cycles decreased from 2010–2050, other impacts, particularly atmospheric aerosol loading and climate change, were expected to increase to different extents. This indicated that trade-offs could transition across the various dimensions of environmental sustainability throughout

wastewater management activities, together with improvements in the practices. Further, Fig. 4b shows that the effects of wastewater management practices are geographically and temporally diverse, which is consistent with the patterns shown in Figs. 1 and 2. In 2010, Asia was the main contributor to all the environmental impacts resulting from domestic wastewater management, with eastern and south-central Asia being of particular concern, followed by the Americas, Africa, Europe, and Oceania (Fig. 4b). However, this situation had changed substantially by 2050, with Africa surpassing Asia to become the main contributor to the nutrient cycle impacts, with western Africa alone accounting for nearly half of the total for the continent. However, by 2050, Asia (particularly eastern Asia) was still the most significant contributor to other broad environmental impacts caused by wastewater treatment services.

## Discussion
Inspired partly by a number of UN actions, global efforts have been focused on developing solutions for countries and regions with little to no sanitation or wastewater infrastructure[25]. Global water security, relative to aquatic eutrophication, has become a significant challenge in recent years, leading to the partial retrofitting of wastewater facilities to remove nutrients beyond the basic functions (degradation of organics and disinfection)[30]. Here, we estimated the broad environmental impacts of incorporating and tightening nutrient removal through the prevalence

of more-sophisticated treatment beyond the impacts associated with primary services. Our results showed that, overall, patterns changed across spatial (national and regional) and temporal scales, in combination with multiple demographic and socio-economic driving factors. As current nutrient impact hotspots related to domestic wastewater management will transition from Asia to Africa by 2050, as shown by our analysis, there is an urgent need to devote more monitoring and intervention efforts to Africa, especially in countries with growing populations, rising daily protein intake, rapid urbanisation, and inadequate sanitation and wastewater facilities.

It has been accepted widely that nutrient removal from domestic wastewater is essential and could have a substantial effect on N cycling[31]. However, our sensitivity analysis illustrated a considerable dilemma, namely, whether concerted efforts should even be devoted towards removing nutrients from domestic wastewater, as such actions could potentially shift the pressures from aquatic ecosystems to the broader environment (particularly climate change and atmospheric aerosol loading). These increased burdens to the atmospheric system might not be comparable with other anthropogenic activities causing climatic variation and air pollution (such as energy and transport)[32,33]. However, the present findings advanced our understanding of how interconnected challenges encountered in a single sector, such as water, should be dealt with from a systems integration perspective. Accordingly, this critical trade-off places a greater focus on innovative wastewater facilities aimed at removing or capturing nutrients from wastewater with minimal or no increase in other types of contamination[34]. Bridging the gap between wastewater treatment services and manufacturing industries is a potential solution, as it would facilitate nutrient recycling beyond nutrient removal alone[35]. Increased nutrient recycling, along with more inherently resource-efficient measures, could possibly reduce their overall environmental impacts[36]. These solutions include source separation (urine diversion[37]) with anaerobic ammonium oxidation (anammox), which reduces energy requirements significantly[38], and enhanced biological P removal integrated with P recovery[39]. In addition to reducing effluent nutrients, such innovations provide opportunities essential to reducing the intensive reliance on commercial fertilisers and other overall impacts rather than simply shifting the pressures across ecosystems[40]. However, research efforts should also be devoted to investigating the economic feasibility of emerging resource recovery paradigms, as conversion costs are crucial constraints in infrastructure transformation[41]. Many environmentally benign options may not be quite cost-effective[42], whereas there might be situations in which nutrient recycling does not reduce environmental impacts[43].

Diets link environmental and human health[44], and our results indicated that significant environmental impacts of wastewater treatment were associated usually with a high level of daily protein consumption (particularly plant-based protein). Therefore, high-protein and plant-based diets could magnify the negative effects of providing wastewater services. However, plant-based diets are considered more environmentally sustainable compared with meat-rich diets because of the lower depletion of resources and reduced carbon emissions[45]. Furthermore, environmental sustainability has to correspond to maintaining healthy nutrition[46]. Interesting studies could therefore be conducted on the connections between dietary habits and wastewater management in the broad context of sustainability. In addition, while per capita dietary protein availability could be peaking in developed countries, it is rising in less-developed countries, where the population and the availability of sanitation will also be increasing. These combined trends could result in significant environmental impacts

in such countries, with less capacity to fund major investments in sanitation infrastructure.

The focus of our study was on the global situation and we assumed optimistic likelihoods of future wastewater management and nutrient removal prevalence to test the dynamics of the environmental response to the growing restrictions on point-source nutrient discharge. However, more-reliable data are needed to deal with the variability in the point-source effluent regulations that are implemented in a given country. Although the UTT regime assumes universal access to efficient nutrient removal facilities, an assumed 10% of N and 5% of P from domestic wastewater would still enter the environment untreated. This suggested that underestimates could occur for developed countries (such as the United States), as more-stringent levels for nutrient removal are in force in the eutrophic waters of such countries. Furthermore, in estimating wastewater nutrient removal, we assumed that a given sanitation coverage implied a certain level of usage. However, we acknowledge that coverage does not always guarantee usage and this assumption is therefore a limitation of our model. Consequently, the model likely overestimated the wastewater volume entering treatment facilities while underestimating the wastewater volume released to the environment without treatment[47]. Moreover, we assumed that all the nutrients from human excreta could enter sanitation and wastewater facilities and receive distinct degrees of treatment and off-site recycling by means of the application of biosolids on land. This model setting might not be true for situations in which sanitation facilities do not discharge effluent streams to receiving waters but could still cause environmental damage (such as groundwater contamination from pit latrines[48]). These assumptions highlighted the need for reliable data on national and subnational scenarios related to sanitation and wastewater services, which would improve the resolution of our integrated model. Additionally, it must be noted that anaerobic sludge digestion, biogas production, and carbon sequestration were not considered in this study because our primary focus was on nutrients. However, incorporating these activities into our model could alter the greenhouse gas emission calculations and the related impacts on the atmospheric environment. In addition to the uncertainties related to model inputs and results, the thresholds obtained from the literature on the planetary boundaries could also involve uncertainties[15]. A detailed analysis of incorporating these potential uncertainties and influences is beyond the scope of this study, as the concept of planetary boundaries was used merely to facilitate understanding of the overall impacts from a global systems perspective. However, this aspect merits further investigation in the future.

Reducing nutrient discharge from point-sources such as domestic wastewater is essential to alleviating the risks of dangerous aquatic eutrophication. The wastewater industry has to strive to improved treatment services, with minimal or no increases in other types of pollution. Recycling nutrients from wastewater to manufacturing and using more inherently sustainable solutions are imperative to reducing nutrients in domestic effluent and the related overall environmental impacts. This study presents information essential to facilitating the engagement of individuals, utilities, and policy-makers in reducing nutrient discharge from domestic wastewater and minimising the related environmental impacts in a truly sustainable manner.

## Methods
**Overview of modelling**. The modelling framework we used to evaluate the broad environmental impacts of reducing global nutrient discharges from domestic wastewater (shown in Supplementary Fig. 1) is described in the following subsections. The quantities of nutrients (N and P) contained in domestic wastewater

were estimated based on the country-level per capita dietary protein intake. The calculations of the environmental impacts associated with point-source nutrients and the implementation of treatments to remove them were based on estimations of the populations served by sanitation and wastewater systems, scaled according to life cycle inventory data (depending on the degree of nutrient removal and the specific treatment technologies applied), and compared with the reported global thresholds for planetary boundaries. To comprehensively estimate the overall impacts arising from wastewater nutrient removal processes, the system boundaries contain both foreground processes (covering aqueous and gaseous emissions throughout wastewater treatment trains) and background processes (including energy and chemical manufacture and the raw materials required to construct and operate wastewater facilities). The facilities needed to produce electricity and chemicals were excluded from the calculations. The main environmental impact categories assessed in this analysis included N and P cycles, climate change, stratospheric ozone depletion, atmospheric aerosol loading, chemical pollution, biodiversity loss, and freshwater use. Uncertainty and sensitivity analyses were performed to assess issues, such as estimation robustness, parameter uncertainty, and the identification of key model inputs.

**Data acquisition and preliminary handling.** Various data sets related to sanitation coverage (including treated and untreated sewer connections and decentralised infrastructure complying with certain standards)[49], dietary protein intake[18], freshwater withdrawal for municipal use[50], and population and urbanisation estimates[16] were collected for 173 countries and territories (collectively referred to as countries in the analysis) in Africa, the Americas, Asia, Oceania, and Europe. The selection of the countries was based on the availability of information and the principle of consistency across the above-mentioned data categories. Overall, the number of selected countries accounted for over 70% of the global total countries, whereas the total populations of these countries comprised 99% of the global population. This ensured the representativeness of this analysis from a global systems perspective (Supplementary Table 2). However, under various circumstances, the temporal ranges of the available country-level values did not match (the source of each data set is presented in Supplementary Table 3), and the quality of the information varied (quality assessment, shown in Supplementary Table 4, was conducted using the evaluation matrix described in Supplementary Table 5). To add consistency across the data sets, only historical values from 1990–2010 were used as inputs in the modelling procedures. In addition, value sets of inferior quality were included in the uncertainty assessment.

To enable projections of future conditions by 2050, population and urbanisation levels were obtained directly from the UN 2014 World Urbanization Prospects[16], whereas historical data on sanitation coverage for the period 1990–2010 were extrapolated linearly to 2050. Any negative slopes were replaced with horizontal slopes for conservativeness, using a procedure similar to that used by Trimmer et al.[34]. This procedure was used also for other country-level data sets for which documented projections were not available. It should be noted that in several countries (marked with asterisks in Supplementary Table 2), improved sanitation facilities are shared between two or more households and the relevant estimates on sanitation coverage we used were obtained directly from the available database[49].

**Wastewater management regimes.** To test the significance of the environmental impacts arising from increased reductions in point-source nutrient discharge, we proposed three distinct wastewater management regimes in three successive time periods from 1990 to 2050 (Supplementary Table 6).

In this study, we used regional estimates and assumptions related to the prevalence of different wastewater treatment levels (primary, secondary, or tertiary treatment), and we subsequently combined them with the nutrient removal efficiencies typically reported for the individual levels (primary treatment: 10% for both N and P; secondary treatment: 40% for N and 45% for P; and tertiary treatment: 90% for N and 95% for P)[12,43,51,52] to simulate the reductions in the nutrient inputs from domestic wastewater. The BAU regime represents regional disparities in point-source nutrient removal practices (in terms of the prevalence of different wastewater treatment facility levels), providing a baseline that represents the actual conditions. The percentages of improved sanitation served by secondary treatment in Africa, the Americas (excluding North America), North America, Asia, Oceania (excluding Australia), Australia, and Europe were approximately 0%, 14%, 90%, 35%, 0%, 90%, and 66%, respectively[22]. The percentages of improved sanitation served by secondary treatment were subtracted from 100% to estimate the proportions of improved sanitation served by primary treatment in these world regions. Country-level or even higher-resolution disparities in regulations to reduce point-source nutrients, and the contextual factors that could potentially influence their implementation, were not considered in this study. This is partly because they are dependent significantly on specific local situations and design systems and partly because of the scarcity of high-resolution data[53]. The STA and UTT regimes represent optimistic likelihoods for future nutrient removal and provide points of comparison with current conditions. We included various technological approaches, individually and in different combinations, to handling waterborne contaminants. These included primary sedimentation, Ludzack–Ettinger, Bardenpho, oxidation ditch, and membrane-based biological methods. A summary of the different treatment approaches is presented in Supplementary Table 7.

In this study, improved sanitation coverage included centralised and decentralised service facilities; therefore, we assumed that the above-mentioned engineering technologies for pollution control were available for both types of wastewater services. Currently, decentralised wastewater treatment is recognised as a possible solution that would satisfy sanitation requirements[54]. In addition, many existing biological approaches used in centralised wastewater facilities could be applied in decentralised systems in small towns, peri-urban areas, and rural communities in developing countries such as China and India[55], depending on the system layout and actual operating conditions[56]. Therefore, the life cycle inventory values related to the engineering-based treatment approaches were assumed to be uniform in modelling the environmental impacts associated with both centralised and decentralised treatment systems. Although nature-based approaches, such as constructed wetlands, are considered potential alternatives to dealing with similar challenges, they were excluded from our analysis. This is because ecological approaches are often relevant to given local conditions and specific system designs[57], resulting in a relatively high level of uncertainty in terms of analysing life cycle impacts[58].

**Model formulation.** To combine various types of information with the wastewater management regimes (for which summaries are provided in Supplementary Table 6), we built a model capable of integrating temporally and spatially variable demographic and socioeconomic drivers to simulate the environmental impacts of alternative water and sanitation services (IDEAS). Building upon the framework of the IDEAS model (version 1.0, Supplementary Fig. 1), the environmental impacts attributed to wastewater management practices can be assessed as follows:

$$P^T(y, G, a) = \sum_{c \in G} P^S(y, c, a) + \sum_{c \in G} P^D(y, c, a) \qquad (1)$$

where $P^T(y,G,a)$ represents environmental impact $a$ in year $y$ from a global perspective, and $G$ covers the 173 countries $c$ listed in Supplementary Table S2. The impact $a$ can represent the N cycle (kg N yr$^{-1}$), P cycle (kg P yr$^{-1}$), climate change (kg $CO_2$-eq yr$^{-1}$), stratospheric ozone depletion (kg CFC11-eq yr$^{-1}$), atmospheric aerosol loading (kg PM10-eq yr$^{-1}$), chemical pollution (disability-adjusted life years — DALYs, yr$^{-1}$), biodiversity loss (species yr$^{-1}$), or freshwater use (m$^3$ yr$^{-1}$). $P^S(y,c,a)$ represents the combined impact of treated effluents (including remaining organic matter, nutrients, and heavy metals), airborne emissions ($CO_2$, $CH_4$, $N_2O$, $NH_3$, SOx, NOx, and CO), and the recycling of solid waste to agriculture (including heavy metals and avoiding the impacts of commercial fertiliser) throughout the entire life cycle of the treatment process, as well as the manufacture of the energy, chemicals, and materials required to operate wastewater systems. $P^D(y,c,a)$ represents the impacts generated from untreated effluent (organics, nutrients, and heavy metals) and concomitant gaseous emissions ($CH_4$ and $N_2O$).

Our model estimated the impacts attributable to the handling of domestic wastewater flows that enter the sanitation and wastewater treatment facilities ($P^S$), and the remaining mass released into the aquatic environment without any treatment ($P^D$), by integrating country-specific and temporal values on the volume of wastewater produced ($M$), the usage of improved sanitation facilities ($\varepsilon$), and life cycle impact operators ($\sigma$), as follows:

$$P^S(y, c, a) = \sum_{j \in \{0,1\}} (M(j, y, c) \cdot \varepsilon(j, y, c)) \cdot \sum_{t \in \{1,2,3\}} (\sigma_t(a) \cdot i_t) \qquad (2)$$

$$P^D(y, c, a) = \sum_{j \in \{0,1\}} (M(j, y, c) \cdot (1 - \varepsilon(j, y, c))) \cdot \sigma_0(a) \qquad (3)$$

where $j$ indicates whether a data set originated from an urban (0) or a rural (1) area, and t whether a data set relates to primary (1), secondary (2), or tertiary (3) wastewater treatment facilities, $i$ represents the prevalence rate of an individual wastewater treatment level $t$, $\sigma_t$ is an operator representing the impacts resulting from wastewater treatment processes, and $\sigma_0$ is an operator representing the impacts attributed to untreated effluent discharge and concomitant gaseous emissions.

To explore the environmental effect ($P^O(y,G,a)$) of wastewater management services from a One-Earth perspective, we built an algorithm relating each environmental impact and its planetary boundary threshold, as follows:

$$P^O(y, G, a) = \frac{P^T(y, G, a)}{B(a)} \qquad (4)$$

where $B(a)$ represents the threshold level of environmental impact $a$.

**Estimating global wastewater production quantities.** We multiplied the country-level estimates of wastewater production rates on a per capita per year basis ($m$, m$^3$ capita$^{-1}$ yr$^{-1}$) by urban and rural population data to compute the volume of wastewater produced ($M$, m$^3$ yr$^{-1}$) in Equations (2) and (3). For this, we estimated the wastewater production rate ($m$) using the following expression:[2]

$$m(j, c) = \omega(c) \cdot (1 - \tau(j)) \qquad (5)$$

where $\omega(c)$ denotes freshwater withdrawal for municipal use (m$^3$ cap$^{-1}$ yr$^{-1}$), and $\tau(j)$ is the water consumption to withdrawal ratio with a median of 0.1 and 0.3 for urban and rural areas, respectively, used in this analysis[59].

 9

**Estimating the usage of sanitation facilities**. We assumed that a given sanitation coverage, including treated and untreated sewer connections, and assuming that decentralised systems achieved certain standards, implied a certain level of usage[49]. For a number of countries (such as Eritrea, Angola, and Libya, as marked by asterisks in Supplementary Table 2), improved sanitation includes facilities shared between two or more households. The rest of the countries use improved facilities not shared with other households.

Furthermore, we assumed that all nutrients from human excreta (regardless of the type of improved sanitation system) had the potential to contaminate the environment, depending on how they were disposed of and the degree of treatment they received (the basis for Equation (2)). Sanitation coverage does not necessarily signify that waste is collected and disposed of at treatment facilities, particularly in some decentralised sanitation systems (e.g. septic tanks and pit latrines). This could be the condition in developing countries and in less-developed regions in a country with inadequate sanitation infrastructure and management[60]. However, currently, the emphasis is on systems approaches to improved sanitation to ensure sustainability[48]. Accordingly, access to sanitation services is considered a multi-step process in which pit latrines or similar decentralised facilities are parts of the chain, to be supported by the collection, transportation, and treatment of waste for safe disposal. Therefore, this study assumed that all the nutrients (N and P) from human excreta could enter sanitation and wastewater facilities. In addition, worldwide, faecal sludge and urine collected from septic tanks and pit latrines are used commonly as fertiliser[61]. Consequently, the environmental burdens and benefits of recycling human excreta (in sewage sludge) to agriculture were included in this study. The nutrients deriving from human excreta from populations without access to improved sanitation systems were also included and were assumed to have received no treatment prior to discharge (Equation (3)). The percentage of the population served by improved sanitation was subtracted from 100% to estimate the proportion of the population not served by improved sanitation facilities. The relevant data for estimates on the coverage of improved sanitation facilities (the model parameter $\varepsilon$ in Equations (2) and (3)) were obtained from the World Health Organization and UN International Children's Emergency Fund Joint Monitoring Programme for Water Supply and Sanitation[49].

**Estimating life cycle impact operators**. The standard life cycle assessment (LCA) conforms to the midpoint and endpoint methods of the prevailing ReCiPe model[62] to consider environmental impact categories. In this study, the LCA results for each treatment approach under the three wastewater management regimes were determined by integrating life cycle inventory data from the literature[63], using a midpoint ReCiPe model in SimaPro (PRé Sustainability, the Netherlands), which is an LCA code. This method was also used to determine the LCA results for situations without wastewater treatment services, using the relevant life cycle inventory data from the same source. Thereafter, the reference LCA results ($\sigma^R$; see Supplementary Table 8) were used to extrapolate the life cycle impact operators ($\sigma$) related to the different wastewater treatment levels:

$$\sigma_t(a) = \sigma^R(a, t) \cdot d(c) \qquad (6)$$

where $d(c)$ is a simplified line extrapolation factor and can be calculated as follows:

$$d(c) = \frac{\frac{L^N(c)}{L^{RE,N}} + \frac{L^P(C)}{L^{RE,P}}}{2} \qquad (7)$$

where $L^{RE,N}$ and $L^{RE,P}$ represent the reference N and P loads in domestic wastewater (50 mg N L$^{-1}$ and 12 mg P L$^{-1}$, respectively)[63]. The domestic wastewater N and P loads in country $c$ were estimated using the following expressions:

$$L^N(c) = \frac{S(c) \cdot \gamma}{\sum_j m(j, c)} \qquad (8)$$

$$L^P(c) = \frac{S^M(c) \cdot \delta^M + S^P(c) \cdot \delta^P}{\sum_j m(j, c)} \qquad (9)$$

where $L^N(c)$ and $L^P(c)$ represent the N and P loads attributed to human excreta in mg N L$^{-1}$ and mg P L$^{-1}$, respectively; $S(c)$, $S^M(c)$, and $S^P(c)$ represent the daily intake values from total protein, meat-based and plant-based protein, respectively, in grams of protein per capita per day; $\gamma$ is a ratio denoting the N content of protein (set to 0.13[24]), and $\delta^M$ and $\delta^P$ are ratios representing the P contents of different protein sources (set to 0.011 for meat- and 0.022 for plant-based protein, respectively[24]). It should be noted that food waste was not considered in this study to estimate dietary protein intake. The LCA model was run several times, assuming the global implementation of a given treatment method each time, while the median value of each impact category for a given nation in a given regime was used to calculate the primary results of this study. The probability distributions of the reference LCA values were used in the uncertainty analysis.

**Estimating planetary boundary thresholds**. The damaging effects of N and P, mainly related to aquatic eutrophication, are usually linked closely. However, LCA studies often use separate emission flows and impacts for N and P, and the references on planetary boundaries typically propose two completely independent boundaries for the N[64] and P[65] cycles. Therefore, our estimates for these flows used the ReCiPe midpoint metrics for marine and freshwater eutrophication, and the

global flows of 62 Tg N yr$^{-1}$ and 11 Tg P yr$^{-1}$ were used as the relevant boundaries[15]. In LCA analysis, the impacts of climate change usually reflect the global warming potential (GWP; kg $CO_2$ equivalent). To include a relatively complete list of greenhouse gases instead of only $CO_2$, an alternative threshold of + 1.0 W m$^{-2}$ was used for climate change[15], and this was converted to 11,507.5 Tg of $CO_2$-eq for easy comparison[66]. The stratospheric ozone depletion potential (kg CFC-11 equivalent) uses the same midpoint method. As the permissible ozone loss is nearly 14 Dobson units (DU)[15], the global allowance of ozone damage is assumed at 14 DU yr$^{-1}$, corresponding to annual emissions of 0.409 Tg CFC11-eq[66]. The midpoint indicator of particulate matter formation (in kg $PM_{10}$ equivalent) used was compared with the global atmospheric aerosol loads (14.57 Tg $PM_{10}$-eq)[67]. The water depletion metric applicable to measuring freshwater use as a planetary boundary is 4000 km$^3$ yr$^{-1}$[15]. The impact of chemical pollution was determined via the human toxicity potential in the endpoint LCA method and expressed as DALYs lost, with the relevant boundary set at 4.36 million DALYs, based on the limited research available[62]. Biodiversity loss was reflected as annual reversible species extinctions and was compared with the global boundary of 280,729 species yr yr$^{-1}$[162,66].

**Uncertainty and sensitivity analysis**. Monte Carlo analysis was conducted to generate input uncertainties and to calculate global ranges for the environmental impact sets of each wastewater management regime. The wastewater treatment approaches were independent of one another (no correlations were expected) and assumptions were included as necessary, with the exception of potential uncertainty in ecoinvent data. The data for each input parameter justified a probability distribution and each input parameter was assigned plausible maximum and minimum values identified in this study (Supplementary Table 8). Owing to the scarcity of data, common in LCA studies, we were unable to capture the true underlying distribution functions of several parameters. Therefore, we used triangular distributions, with a mode and a minimum or maximum, based on the average and limiting values of the available data, respectively. The results (midrange values) presented here were compared with uncertainty values in the fifth and ninety-fifth percentile range from the distribution of the output of 100,000 Monte Carlo simulations run using the IDEAS model (shown in Supplementary Fig. 2). The overall analysis was repeated to confirm that the number of simulations were more than sufficient to generate reproducible results.

To identify the factors that played a significant role in driving trends in the impacts generated by wastewater treatment practices, we selected key model parameters, including demographic, water and food related, and sanitation infrastructure factors (Fig. 3) for one-factor-at-a-time sensitivity analysis. In this instance, we varied each model input in the year 1990 by ± 20%, in turn, observing the effect on the environmental impact categories. A sensitivity index (RSI) was developed relative to the ratio of the change in the output parameter and the change in the input parameter. The global-level RSI results are presented in Fig. 3, with a full version at national scale provided in Supplementary Data 3.

## Data availability

The population statistics and projected trajectories are available from the database of the UN Departments of Economic and Social Affairs (https://esa.un.org/unpd/wup/). The statistics on per capita freshwater withdrawal for municipal use and per capita daily dietary protein intake are available from the UN Food and Agriculture Organization AQUASTAT and FAOSTAT databases (http://www.fao.org), respectively. The data on sanitation coverage can be accessed from the database of the World Health Organization and the UN Children's Fund Joint Monitoring Programme for Water Supply and Sanitation (https://washdata.org/data). All model inputs for the analyses in the present study are available publicly through the literature cited or the data sources provided in the main text and in the Supplementary Information. The main model output from the entire analysis are also provided in the Supplementary Data.

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

## Acknowledgements

We are grateful for the support received from the Beijing Nova Program (Z171100001117078), Beijing Talents Foundation (2017000021223ZK07), National Natural Science Foundation of China (51408589), Youth Innovation Promotion Association of the Chinese Academy of Sciences (2016041), and to the K. C. Wong Education Foundation for sponsoring the Royal Society Newton International Fellowship (NF160404).

## Author contributions

X.W. and D.B. conceived this study; X.W. formulated the model, performed the simulations, analysed the results, prepared the figures, and wrote the paper; X.W., G.D., D.B., W.V., C.K., M.Y., J.L. and N.R. contributed substantially to data interpretation and manuscript editing.

## Additional information

**Competing interests:** The authors declare no competing interests.

