## [Peer Review File · Nature Communications]

Reviewers' comments:

Reviewer #1 (Remarks to the Author):

SUMMARY:

This manuscript develops a modeling framework that integrates various types of information (e.g., population, sanitation coverage, food intake, life cycle inventory data) to evaluate the broad environmental impacts of reducing global nutrient discharges from human-generated wastewater. The work models increasingly effective removal of nitrogen and phosphorus over three successive time periods (1990-2010; 2010-2030; 2030-2050) and estimates the life cycle environmental effects associated with resulting nutrient discharges and the implementation of wastewater facilities capable of achieving these treatment levels. The findings provide information at a country-level resolution, identifying trends and hotspots of high environmental impacts across space and time. Additionally, the study includes a sensitivity analysis to isolate the factors that play the largest role in driving these trends, and it integrates global thresholds from the literature on planetary boundaries to assess the overall global impact of wastewater management.

GENERAL COMMENTS:

Overall, this manuscript presents an interesting analysis that investigates a question important to environmental protection and sustainability, integrating datasets, assumptions, and calculations in a novel way to examine the issue from a global systems perspective. In particular, the sensitivity analysis the authors conduct provides potentially valuable information for national and global decision-makers, as it identifies factors that play a key role in driving these impacts. However, I have some significant questions and concerns related to the study's methodology and interpretation of results. The authors are also encouraged to proofread the manuscript and supplementary materials to check for typographical errors and clarity of writing. The most critical content-related areas the authors should address include:

(i) Providing greater clarity surrounding the datasets, assumptions, and literature sources used as inputs into the model: Supplementary Table S1 provides a summary of the different types and sources of data used, along with the assumptions associated with each dataset. Information is also provided in the Methods section, but, in many cases, additional details, limitations, and clarifications should be provided. This information could be provided in Table S1 and the Methods so all relevant info appears in no additional locations. Additional information that should be provided includes:

- Sources for all data, factors, and assumptions;
- Temporal ranges of all data used from existing datasets;
- The meaning of the phrase "These data sets are assumed to be time-independent", referring to dietary intake and municipal freshwater withdrawal (country-level dietary intake and freshwater withdrawal to change with time, and the referenced datasets provide data over a range of years);
- Methods used if certain data (e.g., dietary intake, municipal freshwater withdrawal) were not available for a given country;
- Whether household-level food waste was considered in relation to FAO dietary intake (the FAO dataset does not account for household waste);
- How the different treatment methods (A1-A5) were included in the analysis (e.g., did the model run multiple times, assuming global implementation of a given treatment method each time, or did it include an algorithm to select different methods within a single run?);
- The fact that the ratio between protein intake and P emissions is dependent upon whether the protein source is vegetable-based or meat-based (Equation 9 in the Methods implies this point, but it is not stated explicitly in the text or Table S1);
- Whether improved sanitation coverage (used here as a surrogate for sanitation usage) included systems shared by more than one household (now classified as "limited") or systems used by a single household (now classified as "basic");
- The fact that improved sanitation coverage does not necessarily imply a wastewater discharge and connection to wastewater collection and treatment facilities (e.g., pit latrines and septic

systems do not discharge nutrients to surface waters)

(ii) Presenting the results of the uncertainty analysis: The authors refer to a Monte Carlo analysis with 100,000 simulations to assess possible variations in model inputs. The Monte Carlo analysis is presented as being distinct from the extended scenarios included in the sensitivity analysis (as it should be), but the Monte Carlo results are not shown or discussed anywhere in the manuscript. Additionally, neither the procedures used for defining the uncertainty distributions associated with each model input nor the distributions themselves are presented anywhere in the manuscript or supplementary material. The uncertainty analysis constitutes an important piece of this work, as a number of the datasets and assumptions used in the overall model may be associated with substantial uncertainty or variability. The authors should include more information surrounding the Monte Carlo analysis's implementation and findings.

(iii) Reframing the Results and Discussion to focus on each of the key drivers identified in the sensitivity analysis: One of the strengths of this work is the fairly extensive sensitivity analysis conducted to identify key drivers associated with changing environmental impacts. These identified factors (e.g., population, sanitation coverage, protein intake) could provide an interesting framing device when discussing and interpreting results. As it stands now, the sensitivity analysis is discussed near the end of the Results section, whereas an alternative structure that highlights each of the critical factors separately might emphasize how individuals, utilities, and policy-makers could help to minimize the environmental impacts associated with nutrient discharges and their reduction. To be clear, I am suggesting the authors structure the Results to include a separate section for each of the 3-4 most critical factors (population, sanitation, protein intake, etc.), where they discuss why it is important, how it contributes to environmental impacts, the countries in which it plays the most crucial roles, and how this information might be used to minimize potential increases in negative environmental effects. This would require a fairly significant rewrite of much of the paper and potentially the inclusion of different figures (perhaps in place of Figure 2) to complement this new structure. However, I feel it would enhance key messages regarding how the model's results can be used.

(iv) Revising the "one-Earth perspective" analysis, especially to move away from aggregating all environmental effects: The final aspect of the analysis (discussed in the last two paragraphs of the Results section and presented in Figure 3) represents an interesting application of the planetary boundaries, comparing the environmental effects of nutrient removal from wastewater with Earth system thresholds estimated in previous literature. First, it is important for the authors to clearly state that the planetary boundaries represent estimated thresholds for all human activity, not just wastewater treatment. Wastewater treatment is important, but it is only a small piece of the overall impact of humanity. Even if the calculations show nutrient removal from wastewater to be well within the planetary boundaries, that does not mean it is sustainable. It may still have an outsized impact relative to other human activities. Second, some of the assumptions implicit in this discussion and in Figure 3b are not necessarily valid. In particular, aggregating all environmental effect categories together (by summing normalized values, as appears to be the case in Figure 3, or by averaging, as is described in the Methods) assumes that each effect category has an equivalent impact on the overall Earth system (relative to the estimated threshold values). This is not the case. Additionally, aggregating the effects suggests that wastewater treatment has a relatively consistent impact (relative to humanity's overall impact) across all categories, which is also not the case. For example, it may be particularly significant with regard to nutrient cycles, whereas it may be relatively insignificant for ozone depletion. The authors should avoid making these assumptions. I would suggest revising Figure 3b and any accompanying text to focus on a more qualitative discussion around tradeoffs across different effect categories.

SPECIFIC COMMENTS:

Line 48. Typographical error: "tonne" should be "tonnes".

Lines 54-56. "...the amounts of N or P passing through human metabolism to wastewater are comparable with the key aspects of the global N or P cycle." This statement is unclear. The authors should specify to which aspects of global nutrient cycles they are referring.

Lines 61-63. Though I think I understand the meaning of this sentence, I am finding the wording to be confusing. Here is an alternative option: "Yet, the wastewater industry has placed less emphasis on broader environmental responses to the growing number of nutrient discharge interventions."

Lines 64-66. This sentence may fit better in the preceding paragraph, while the last sentence of that preceding paragraph may align more closely with the remainder of this current paragraph. The authors are encouraged to consider switching the order of this sentence and the preceding one.

Line 78. The meaning of "these" is unclear. The authors should clarify whether they are referring to environmental consequences, restrictions on nutrient discharges, or synergies and tradeoffs across environmental effects. Additionally, this sentence could be revised to be much stronger so that it drives home the overarching objective of the work, which I don't believe is ever explicitly stated in the manuscript.

Line 83. The word "analysis" should be "analyses".

Line 86. The phrase "thresholds of the Earth systems" is somewhat unclear. The authors should specify that they are referring to the thresholds defined in the planetary boundaries framework and include citations.

Line 90. Referring to the three time periods with increasing nutrient removal regimes (1990-2010; 2010-2030; 2030-2050) as "scenarios" does not quite fit. To me, "scenarios" imply alternative possible trajectories modeled to compare outcomes within a specified spatial and temporal setting. This study models three distinct time periods, each with a different treatment regime. Here and throughout the manuscript, the authors should consider changing this word to "periods", "regimes", or something similar.

Lines 91-92. The authors acknowledge this limitation later, but the use of globally uniform nutrient treatment efficiencies may be an oversimplification, especially for the historical time period (1990-2010). Other studies (e.g., Moree et al., 2013; Van Drecht et al., 2009) have evaluated historical and future nutrient discharges, at least for urban areas, and have differentiated nutrient treatment levels within and across countries. The authors should either provide further justification for using globally uniform treatment efficiencies or consider developing more refined estimates differentiated by country or region. Additionally, it would be informative to compare nutrient discharge estimates from this study with those developed in previous studies.

Line 128. Remove the word "remarkable", and any other subjective language like it.

Lines 128-162. Many of the trends described in these paragraphs are not all that surprising, predominantly following historical and projected population trends. As the sensitivity analysis shows, the substantial variations seen across these countries can often be explained in a fairly straightforward manner, as they are highly associated with drivers such as population, sanitation coverage, and protein intake. Beginning here, the alternative structure described in the General Comments (using the sensitivity analysis and the key drivers it identified as a framing device) would explicitly address and help to explain these trends.

Lines 160-162. I am unclear as to why the model would underestimate discharges where open defecation is practiced. If I am interpreting the Methods section correctly, it suggests that

nutrients from populations without improved sanitation systems are also included and are assumed to receive no treatment prior to discharge (Equation 3).

Lines 171-174. This sentence should include some explanation as to why these locations could see co-benefits. Alternatively, since Figure 2 provides the basis for this statement, is it possible that the environmental effects in Europe and the Caribbean are only declining relative to places like Africa, where effects are likely increasing substantially? In other words, perhaps the observed relative decline across multiple effects is not due to absolute decreases in magnitude. The authors should clarify whether or not this is the case.

Lines 204-206. In the extended (sensitivity) scenarios concerning population, urbanization, and sanitation in the year 2050, was only one of these parameters changed to the projected 2050 level while holding the other parameters constant at their 1990 levels? If so, why not hold everything at a given year (say, 2050), and then change one of the parameters by a set percentage? While the authors do take this general approach, the use of 1990 as a baseline while one parameter is changed to its 2050 level is somewhat confusing and counterintuitive. It is not necessarily wrong, but the authors should be clear about this point to reduce any potential confusion.

Lines 246-250. These statements concerning the relative share of total effects from individual effects make the assumption that all eight planetary boundaries have equivalent impacts on the overall Earth system (discussed above in General Comments). This assumption should be avoided, and the authors are encouraged to revise this discussion to focus on more qualitative tradeoffs across different effect categories.

Lines 255-257. This statement should have a citation.

Line 265. "obsolete sanitation facilities". Does this phrase refer to countries with low sanitation coverage, or to something different? The authors should clarify this point.

Lines 272-280. While the focus of this study is on environmental impacts of nutrient removal, it would be worthwhile to mention some of the additional tradeoffs associated with treatment cost and resource recovery here. The most environmentally benign options may not be the most cost-effective, while there may be situations where nutrient recycling does not reduce environmental impacts.

Lines 286-290. I appreciate that the authors acknowledge the limitation that sanitation coverage does not necessarily imply proper sanitation usage. It is also important to note that sanitation coverage does not always imply connection to a sewer wastewater system. Improved sanitation facilities can include options such as septic systems, pit latrines, and decentralized resource recovery facilities, which typically do not discharge effluent streams to receiving waters. They can still cause environmental damage (e.g., groundwater contamination from pit latrines), but those impacts would require sets of assumptions that are wholly different from the universal assumptions made in this study around centralized treatment systems.

Lines 317-322. While Supplementary Table S1 is referenced in the subsequent section, it would be helpful to mention it here as well. Additionally, I would avoid phrasing that states data from more than 60 years (1990-2050) are used. Any projected values (i.e., those going beyond historical datasets) should not be labeled as "data". I would classify any initial inputs to the overall model as "data", and I would refer to any projected values as "projections" or "estimates".

Lines 335-337. While Supplementary Database S3 is referenced at the end of this section, it would be helpful to mention Table 1 here.

Line 363. The word "estimate" should be "estimates".

Line 366. I am hesitant to call these data “geospatial”. While they do provide geographic diversity from country to country, “geospatial” often refers to data at a finer spatial resolution. I would prefer the term “country-level” or something similar.

Line 423. The word “liked” should be “linked”.

Line 426-428. It is worth noting that, in addition to the uncertainty surrounding the model inputs and results, the planetary boundaries are also associated with substantial zones of uncertainty. In particular, Steffen et al. (2015) report the range for the boundary around global phosphorus discharges to water as being 11-100 Tg P per year.

Figure 1. Instead of displaying points with different colors and sizes to represent N and P discharges, the authors might consider including two columns of maps, one for N and one for P, with the countries shaded according to discharge levels. This format might be clearer and make trends easier for the reader to see. In line with my General Comment concerning reframing the results, it might be appropriate to normalize the masses of discharged nutrients relative to population, to account for wide population differences across countries.

Figure 2. The authors might consider switching the color scale so that red is larger than yellow. I think that would be more intuitive for readers. Also, again, reframing the results around key drivers of these discharges may lead to a different figure, or a similar one showing slightly different information (e.g., environmental effects relative to population).

Reviewer #2 (Remarks to the Author):

The manuscript NCOMMS-18-13690-T presents an interesting study on the impact of nutrient (nitrogen and phosphorous) removal from wastewater into different LCA impact categories. The study has a worldwide perspective and considers both spatial and temporal variability. Thus, it allows to identify the change of trends on the global hotspots, i.e. where (geographically) to focus efforts in the incoming decades.

The paper is well written and both the aim and structure are clear. However, if the following issues are considered before publication, they would strengthen significantly the conclusions drawn and the interest and impact of the publication.

Main concerns:

- Calibration of the model presented is not mentioned in the paper. All results presented and conclusions withdrawn would be more reliable if some kind of calibration could be made. For example, could any of the modelled estimations of N and P load be compared with real nutrient loads, maybe for some countries or for some regions?
- A reference for the IDEAS model should be provided.
- The title of the paper refers to the impacts of nutrients discharge reduction for 'ecosystem' sustainability but, indeed, ecosystem understood as receiving waters is not considered; the model does not take into account the impacted waters but of course it is not the same to discharge into a receiving media with high or low river flow rates or to discharge into pristine or heavily polluted waters; how could this be taken into account?
- Only point source pollution due to domestic wastewater is considered, why not industrial wastewater? (even though authors recognize its importance for example in page 14, end of the discussion). And industrial wastewater impact can have special relevance in developing countries where legislation limits are not so strict yet.
- Scenarios: why the 3 initial scenarios (I, II and III) only considers increasing removal efficiencies for N and P but, for example, do not consider an increase on the sanitation coverage (i.e. percentage of people whose wastewater is being treated), even though it is probably a more

realistic scenario? Specially in views of the MDGs.

- Indeed the scenarios are not very well described, they are a bit confusing; page 16 mentions 18 scenarios, where can they be seen? their description can be improved. Also Table 1 is very difficult to understand.

- Scenario 'Stringent nutrient removal' in Table 1 is not Scenario III? Why freshwater use in this scenarios experiences a so huge increase (+202%) ?

- Also, why only 'conventional hard-tech' treatment solutions (i.e. MLE, MBR. etc in Table S5) have been considered? Why nature-based solutions (NBS, e.g. constructed wetlands or stabilization ponds), which can be very well fitted in some developing regions, but also in developed regions as a polishing additional treatment step, have not been considered? Operation of this type of NBS offer some advantages with respect to conventional treatment solutions (in terms of energy and chemicals use, maintenance needed, etc.)

- Regarding the future scenarios defined for the sensitivity analysis, why scenarios with renewable energy and/or water reuse and recycling and nitrogen and phosphorous recovery (instead of consuming energy and chemicals for nutrient removal) have not been considered? Nowadays and all over the world, with the circular economy approach and with already with some full scale implementations, it is not so unrealistic to consider resource recovery (instead of nutrient removal) for some parts of world, at least partially. Indeed this is mentioned in the discussion section, pages 12-13, so why not considered as a potential scenario? Also, optimisation of 'conventional' treatment technologies will reduce GHG emissions (this has been a research hot topic for many years, so one can also expect operation of existing treatment systems or new treatment systems reducing GHGs emissions.

- LCI: how CH₄ and N₂O have been estimated? (some published emission factors produce extremely high -so unrealistic- estimated values)

- LCI: Why anaerobic digestion and biogas uptake are not considered in Table S5?

- LCI: Table S6a includes all construction materials needed for the 5 treatment technologies considered; however, does the LCA study really includes construction of the treatment plants? When reading the manuscript, it seems only operation is considered (for example, lines 355-362 page 17, model formulation section); anyway, if considered, why Kg or Tons of used materials are increasing with scenarios moving from I to III?

- Also, in many parts of the world eutrophication is largely influenced by diffuse pollution, any idea on how to take into account?

- Why only 173 countries have been considered? which have been the selection criteria? are these countries equally distributed all over the world?

- Fig. 2: regional and temporal variations are very difficult to see; might be better to zoom into a specific region where there is a clear change (e.g. Africa) and another one where there is little change (e.g. Europe)?

- page 8, lines 171-174: "...co-benefits could be achieved in most countries from Europe an Caribbean..." how do you see that? how this has been considered?

- In order to estimate wastewater production, wouldn't it be better to use P.E. instead of inhabitants? which other commercial or industrial activities are considered in the equation 5? or is it really only domestic wastewater?

- Equation 7 is using LRE,N and LRE,P but text (line 409) refers to LR,N and LR,P.

- In short: reduction of nutrient loads causes a reduction in eutrophication but an increase in climate change (CC) and aerosol loadings but, could the authors put these increase in perspective/proportion with respect to other activities causing CC or atmospheric pollution, for example future scenarios of transport or mobility or future energy production ? aren't the latter much more relevant to CC than wastewater treatment?

-Oceania's situation is not mentioned in the introduction (page 3)

Response to Reviewer 1

1. This paper presents an interesting analysis that investigates a question important to environmental protection and sustainability, integrating datasets, assumptions, and calculations in a novel way to examine the issue from a global systems perspective. In particular, the sensitivity analysis the authors conduct provides potentially valuable information for national and global decision-makers, as it identifies factors that play an important role in driving these impacts. However, I have some significant questions and concerns related to the study's methodology and interpretation of results. The authors are also encouraged to proofread the manuscript and supplementary materials to check for typographical errors and clarity of writing.

R: We appreciate your commendation of the significance and novelty of our work. In addition, we highly appreciate the comments your raised, as you have not only helped us improve our manuscript but have also provided interesting ideas for future studies. We have made substantial revisions to the manuscript and provide our point-by-point response below.

2. Providing greater clarity surrounding the data sets, assumptions, and sources used as inputs into the model: Supplementary Table S1 provides a summary of the different types and sources of data used, along with the assumptions associated with each dataset. Information is also provided in the Methods section, but, in many cases, additional details, limitations, and clarifications should be provided. This information could be given in Supplementary Table S1 and the Methods, so all relevant information appears in no additional locations. Additional information that should be provided includes:

- (1) Sources for all data, factors, and assumptions.
- (2) Temporal ranges of all data used from existing datasets.
- (3) The meaning of the phrase "These data sets are assumed to be time-independent", referring to dietary intake and municipal freshwater withdrawal (country-level dietary intake and freshwater withdrawal to change with time, and the referenced datasets provide data over a range of years).
- (4) Methods used if certain data (e.g., dietary intake, municipal freshwater withdrawal) were not available for a given country.
- (5) Whether household-level food waste was considered in relation to FAO dietary intake (the FAO dataset does not account for household waste).
- (6) How the different treatment methods (A1-A5) were included in the analysis (e.g., did the model run multiple times, assuming global implementation of a given

treatment method each time, or did it include an algorithm to select different methods within a single run?).

- (7) The fact that the ratio between protein intake and P emissions is dependent upon whether the protein source is vegetable-based or meat-based (Equation 9 in the Methods implies this point, but it is not stated explicitly in the text or Table S1).
- (8) Whether improved sanitation coverage (used as a surrogate for sanitation usage) included systems shared by more than one household (now classified as “limited”) or systems used by a single household (now classified as “basic”).
- (9) The fact that improved sanitation coverage does not necessarily imply a wastewater discharge and connection to wastewater collection and treatment facilities (e.g., pit latrines and septic systems do not discharge nutrients to surface waters).

R: We accepted these suggestions and have made appropriate revisions to provide additional information for improved clarity.

- (1) The entire manuscript has been improved substantially to include sources from the literature for all data, factors, and assumptions. We have modified the Methods section, adding several new subsections in the main text and adding information to Supplementary Tables S2–S12.
- (2) The temporal ranges of all data sets have been provided in Supplementary Table S3.
- (3) We have rewritten these statements to add clarity to the revised summary of the wastewater management regimes (Supplementary Table S7).
- (4) The 173 countries were chosen based on the availability of all the data needed for this analysis and the principle of consistency across data categories. Therefore, the country-level data sets included in the present analysis are available for all the 173 countries listed in Supplementary Table S2. These statements have been added to the revised text (Pages 21–22, Lines 456–472).
- (5) Household level food waste was not considered in this analysis.
- (6) Herein the LCA model was run multiple times, assuming global implementation of a given treatment method each time, while the average value of each environmental impact category for a given country under a given regime was used to calculate the primary results of this analysis. The probability distributions of the data sets were used in the subsequent analysis. The relevant descriptions have been added to the revised text for clarity (Page 30, Lines 654–658).
- (7) The main text has been modified so that this is presented explicitly (Page 11, Lines 229–231; Page 13, Lines 269–274; Page 30, Lines 649–654).

- (8) For a few of the countries considered (such as Eritrea, Angola, and Libya, marked with asterisks in the new Supplementary Table S2), improved sanitation coverage includes facilities shared between two or more households. In the rest of the countries, improved facilities that are not shared between households. We have updated the revised text accordingly (Page 27–28, Lines 594–598).
- (9) In this study, we assume that nutrients from human excreta (no matter what type of improved sanitation system they enter) have the potential to contaminate the environment, depending on how they disposed of further and the degree of treatment they receive. We acknowledge that improved sanitation coverage does not necessarily signify the implementation of waste collection and subsequent disposal, particularly for some decentralized sanitation systems (e.g. septic tanks and pit latrines). This can be the case in developing countries and less-developed regions within a country with inadequate sanitation infrastructure and management. However, a systems approach to improved sanitation is currently being emphasized to ensure the sustainability of the improvements. In this case, the provision of access to sanitation services is considered a multi-step process, in which pit latrines or other similar decentralized facilities are parts of the chain and are supported by the collection and transportation of waste as well as its treatment for safe disposal. Therefore, we assume that all anthropogenic nutrients could enter sanitation and wastewater facilities. In addition, we include a variation in the analysis to account for the fact that using faecal sludge and urine collected from septic tanks and pit latrines as fertilizers is a common practice in many parts of the world. Thus, the environmental burdens and benefits of recycling human excreta waste (in terms of sewage sludge) to agriculture are also included in this study. We have modified the relevant sections in the main text to clarify the basis of our assumptions on this point (Pages 28–29, Lines 599–625).

3. Presenting the results of the uncertainty analysis: The authors refer to a Monte Carlo analysis with 100,000 simulations to assess possible variations in model inputs. The Monte Carlo analysis is presented as being distinct from the extended scenarios included in the sensitivity analysis (as it should be), but the Monte Carlo results are not shown or discussed anywhere in the manuscript. Additionally, neither the procedures used for defining the uncertainty distributions associated with each model input nor the distributions themselves are presented anywhere in the manuscript or supplementary material. The uncertainty analysis constitutes an important piece of this work, as a number of the datasets and assumptions used in the overall model may be associated with substantial uncertainty or variability. The authors should include more information surrounding the Monte Carlo analysis's implementation and

findings.

R: We considered these comments carefully. We conducted a Monte Carlo analysis to propagate input uncertainties mainly for different wastewater treatment approaches and to quantify global averages of life cycle environmental impacts for each wastewater management regime. In the revised main text, we added a new section to the Methods section (Pages 31–32, Lines 684–700) regarding the implementation of uncertainty analysis. We also added information related to the probability distributions of the model parameters to Supplementary Table S12. In addition, the primary results of this analysis were compared with uncertainty values ranging from the 5th to 95th percentile of the distribution of the outputs from 100,000 Monte Carlo simulations. The overall analysis was repeated to confirm that this number of simulations were more than sufficient to generate reproducible results. Building upon the uncertainty analyses (Supplementary Figure S2), we demonstrated that the findings in this study are robust in terms of the distinct assumptions of different wastewater treatment approaches included in each wastewater management regime.

4. Reframing the Results and Discussion to focus on each of the key drivers identified in the sensitivity analysis: One of the strengths of this work is the fairly extensive sensitivity analysis conducted to identify key drivers associated with changing environmental impacts. These identified factors (e.g., population, sanitation coverage, protein intake) could provide an interesting framing device when discussing and interpreting results. As it stands now, the sensitivity analysis is discussed near the end of the Results section, whereas an alternative structure that highlights each of the critical factors separately might emphasize how individuals, utilities, and policy-makers could help to minimize the environmental impacts associated with nutrient discharges and their reduction. To be clear, I am suggesting the authors structure the Results to include a separate section for each of the 3-4 most critical factors (population, sanitation, protein intake, etc.), where they discuss why it is important, how it contributes to environmental impacts, the countries in which it plays the most crucial roles, and how this information might be used to minimize potential increases in negative environmental effects. This would require a fairly significant rewrite of much of the paper and potentially the inclusion of different figures (perhaps in place of Figure 2) to complement this new structure. However, I feel it would enhance key messages regarding how the model's results can be used.

R: The goal of this work was to integrate various demographic and socioeconomic factors with three distinct wastewater management regimes to evaluate the broad environmental impacts of reducing global nutrient discharges from anthropogenic wastewater. Currently, many investments in the wastewater industry have been made

to meet the regulations restricting nutrient discharges. However, it remains unclear that whether and how wastewater management affects the regulation of biogeochemical processes in the biosphere and ensures sustainability. Addressing these issues requires a systems-based approach coupling various human and natural components that interact across technologies, space, and time; this would allow an understanding of the interconnectivity and complexity of the entire system. To this end, we aimed to develop a modelling approach capable of integrating different types of country-level data (e.g., population, urbanization, food and water supplies) that was highly relevant to sanitation and wastewater services. Building on the model, we first explored the magnitude of changes in nutrient flows based on enhanced wastewater services and found that the development of wastewater systems and enhanced nutrient removal practices caused large-scale declines in nutrient inputs to surface waters. We observed that these benefits are unfortunately accompanied by unexpected detrimental effects to broader natural ecosystems. These trends might be attributable to a combined effect of demographic and socioeconomic drivers. Thus, we conducted sensitivity analyses of one factor at a time to pinpoint the ones that played a relatively significant role in driving these impacts. Finally, we integrated global thresholds from the literature on planetary boundaries to help us understand the overall global impact of wastewater management trajectories. Accordingly, we believe that this thought has been developed in a clearer way and that is more understandable for readers. Nevertheless, we highly appreciate your thoughtful and constructive suggestions, and have clarified this in the revised manuscript (Pages 4–5, Lines 78–91). In addition, we have greatly improved the part in the discussion section regarding the findings of the sensitivity analysis, focusing on the key drivers in separated (Pages 10–14, Lines 218–301).

5. Revising the “one-Earth perspective” analysis, especially to move away from aggregating all environmental effects: The final aspect of the analysis (discussed in the last two paragraphs of the Results section and presented in Figure 3) represents an interesting application of the planetary boundaries, comparing the environmental effects of nutrient removal from wastewater with Earth system thresholds estimated in previous literature. First, it is important for the authors to clearly state that the planetary boundaries represent estimated thresholds for all human activity, not just wastewater treatment. Wastewater treatment is important, but it is only a small piece of the overall impact of humanity. Even if the calculations show nutrient removal from wastewater to be well within the planetary boundaries, that does not mean it is sustainable. It may still have an outsized impact relative to other human activities. Second, some of the assumptions implicit in this discussion and in Figure 3b are not necessarily valid. In particular, aggregating all environmental effect categories

together (by summing normalized values, as appears to be the case in Figure 3, or by averaging, as is described in the Methods) assumes that each effect category has an equivalent impact on the overall Earth system (relative to the estimated threshold values). This is not the case. Additionally, aggregating the effects suggests that wastewater treatment has a relatively consistent impact (relative to humanity's overall impact) across all categories, which is also not the case. For example, it may be particularly significant with regard to nutrient cycles, whereas it may be relatively insignificant for ozone depletion. The authors should avoid making these assumptions. I would suggest revising Figure 3b and any accompanying text to focus on a more qualitative discussion around tradeoffs across different effect categories.

R: We have taken these comments seriously and have revised both Figure 3 and the relevant analyses to remove the section relating to the algorithmic aggregation of all the environmental effects. Moreover, we have also improved our relevant analyses to avoid the inappropriate assumptions you raised, concentrating on to a more qualitative discussion surrounding the tradeoffs across different impact categories (Pages 14–15, Lines 303–342; Page 17, Lines 360–369).

6. Line 48. Typographical error: “tonne” should be “tonnes”.

R: The word ‘tonne’ has been replaced with ‘tonnes’ (Page 3, Line 47).

7. Lines 54-56. “...the amounts of N or P passing through human metabolism to wastewater are comparable with the key aspects of the global N or P cycle.” This statement is unclear. The authors should specify to which aspects of global nutrient cycles they are referring.

R: Thank you for this comment. The relevant statement has been rewritten for added clarity in the revised manuscript (Page 3, Lines 53–56).

8. Lines 61-63. Though I think I understand the meaning of this sentence, I am finding the wording to be confusing. Here is an alternative option: “Yet, the wastewater industry has placed less emphasis on broader environmental responses to the growing number of nutrient discharge interventions.”

R: We have modified this sentence (Pages 3–4, Lines 66–67).

9. Lines 64-66. This sentence may fit better in the preceding paragraph, while the last sentence of that preceding paragraph may align more closely with the remainder of this current paragraph. The authors are encouraged to consider switching the order of this sentence and the preceding one.

R: We appreciate this suggestion and have switched the order of these two sentences (Page 3–4, Lines 62–67).

10. Line 78. The meaning of “these” is unclear. The authors should clarify whether they are referring to environmental consequences, restrictions on nutrient discharges, or synergies and tradeoffs across environmental effects. Additionally, this sentence could be revised to be much stronger so that it drives home the overarching objective of the work, which I don’t believe is ever explicitly stated in the manuscript.

R: We accept your concerns and have rewritten this sentence to address both of them (Pages 4–5, Lines 78–91).

11. Line 83. The word “analysis” should be “analyses”.

R: The word ‘analysis’ has been replaced by ‘analyses’ (Page 4, Line 86).

12. Line 86. The phrase “thresholds of the Earth systems” is somewhat unclear. The authors should specify that they are referring to the thresholds defined in the planetary boundaries framework and include citations.

R: We accept this suggestion and have made changes to the relevant phrase (Page 5, Lines 89–91).

13. Line 90. Referring to the three time periods with increasing nutrient removal regimes (1990-2010; 2010-2030; 2030-2050) as “scenarios” does not quite fit. To me, “scenarios” imply alternative possible trajectories modeled to compare outcomes within a specified spatial and temporal setting. This study models three distinct time periods, each with a different treatment regime. Here and throughout the manuscript, the authors should consider changing this word to “periods”, “regimes”, or something similar.

R: We have considered this suggestion carefully and have changed the word “scenarios” to “regimes”, in this context, throughout the revised manuscript.

14. Lines 91-92. The authors acknowledge this limitation later, but the use of globally uniform nutrient treatment efficiencies may be an oversimplification, especially for the historical time period (1990-2010). Other studies (e.g., Moree et al., 2013; Van Drecht et al., 2009) have evaluated historical and future nutrient discharges, at least for urban areas, and have differentiated nutrient treatment levels within and across countries. The authors should either provide further justification for using globally uniform treatment efficiencies or consider developing more refined estimates differentiated by country or region. Additionally, it would be informative to compare

nutrient discharge estimates from this study with those developed in previous studies.

R: We appreciate your comment and suggestion. In the revised text, we have provided extended justification for the use of globally uniform treatment efficiencies in the three wastewater management regimes (Pages 23–25, Lines 489–543). In addition, we have provided comparative analyses of our nutrient discharge estimates with those developed in previous studies in the revised text (Pages 8–9, Lines 161–192).

15. Line 128. Remove the word “remarkable”, and any other subjective language like it.

R: The word “remarkable” has been removed from the text and other subjective language has been avoided in the revised text.

16. Lines 128-162. Many of the trends described in these paragraphs are not all that surprising, predominantly following historical and projected population trends. As the sensitivity analysis shows, the substantial variations seen across these countries can often be explained in a fairly straightforward manner, as they are highly associated with drivers such as population, sanitation coverage, and protein intake. Beginning here, the alternative structure described in the General Comments (using the sensitivity analysis and the key drivers it identified as a framing device) would explicitly address and help to explain these trends.

R: As stated in response to Comment 4, we still prefer to use our original method of framing and our findings in the revised manuscript. Nevertheless, we appreciate your suggestions, and we have rewritten the relevant paragraphs to clarify our points further (Pages 10–14, Lines 218–301).

17. Lines 160-162. I am unclear as to why the model would underestimate discharges where open defecation is practiced. If I am interpreting the Methods section correctly, it suggests that nutrients from populations without improved sanitation systems are also included and are assumed to receive no treatment prior to discharge (Equation 3).

R: We apologize for including the statements that you pointed out. We have removed these from the text.

18. Lines 171-174. This sentence should include some explanation as to why these locations could see co-benefits. Alternatively, since Figure 2 provides the basis for this statement, is it possible that the environmental effects in Europe and the Caribbean are only declining relative to places like Africa, where effects are likely increasing substantially? In other words, perhaps the observed relative decline across multiple effects is not due to absolute decreases in magnitude. The authors should

clarify whether or not this is the case.

R: We appreciate these comments and suggested solutions for added clarity. We have reworked the relevant paragraphs in the revised text (Pages 9–10, Lines 194–212), and provided analyses and findings in the Supplementary Table S1.

19. Lines 204–206. In the extended (sensitivity) scenarios concerning population, urbanization, and sanitation in the year 2050, was only one of these parameters changed to the projected 2050 level while holding the other parameters constant at their 1990 levels? If so, why not hold everything at a given year (say, 2050), and then change one of the parameters by a set percentage? While the authors do take this general approach, the use of 1990 as a baseline while one parameter is changed to its 2050 level is somewhat confusing and counterintuitive. It is not necessarily wrong, but the authors should be clear about this point to reduce any potential confusion.

R: One aim of the present study was to pinpoint factors that play a fairly significant role in driving trends in the environmental impacts of reducing the nutrient content of wastewater. We used sensitivity analyses to explore this. Each of the model factors (e.g. population, urbanization, food intake, and sanitation usage) has a particular significance in terms of presenting a key component of the demographic and socioeconomic systems, which are highly relevant to country level and temporal variations. Thus, we used the status of these factors in 1990 as a baseline while changing only one factor to its 2050 level, performing the sensitivity analysis of one factor at a time. Implementing this procedure helped us better explore how the impacts changed once a key demographic and socioeconomic component developed up to a given status (in terms of a given year) while the other components remained at the baseline year level. Nevertheless, we appreciate your comment and have improved the relevant statements to add clarity regarding the sensitivity analysis we conducted (Pages 32–33, Lines 701–725).

20. Lines 246–250. These statements concerning the relative share of total effects from individual effects make the assumption that all eight planetary boundaries have equivalent impacts on the overall Earth system (discussed above in General Comments). This assumption should be avoided, and the authors are encouraged to revise this discussion to focus on more qualitative tradeoffs across different effect categories.

R: We accept this suggestion and have reconstructed the relevant analyses to focus on more discussing the qualitative tradeoffs across different impact categories (Pages 14–15, Lines 303–342; Pages 17–18, Lines 360–385).

21. Lines 255-257. This statement should have a citation.

R: We accept this suggestion and have added a citation to the relevant statement in the revised text (Page 16, Lines 347–350).

22. Line 265. “obsolete sanitation facilities”. Does this phrase refer to countries with low sanitation coverage, or to something different? The authors should clarify this point.

R: We appreciate the comment and we have rewritten the relevant statements (Page 17, Lines 358–359).

23. Lines 272-280. While the focus of this study is on environmental impacts of nutrient removal, it would be worthwhile to mention some of the additional tradeoffs associated with treatment cost and resource recovery here. The most environmentally benign options may not be the most cost-effective, while there may be situations where nutrient recycling does not reduce environmental impacts.

R: We consider these suggestions seriously and have added a discussion of this in response to the concerns you raised (Page 18, Lines 381–385).

24. Lines 286-290. I appreciate that the authors acknowledge the limitation that sanitation coverage does not necessarily imply proper sanitation usage. It is also important to note that sanitation coverage does not always imply connection to a sewerage wastewater system. Improved sanitation facilities can include options such as septic systems, pit latrines, and decentralized resource recovery facilities, which typically do not discharge effluent streams to receiving waters. They can still cause environmental damage (e.g., groundwater contamination from pit latrines), but those impacts would require sets of assumptions that are wholly different from the universal assumptions made in this study around centralized treatment systems.

R: We considered these comments seriously and have provided extended justification for the relevant assumptions developed in our analysis. We assume that nutrients from human excreta nutrients (no matter what type of improved sanitation system they enter) have the fair potential to contaminate the environment, depending on how they are disposed of further and the degree of treatment they receive (basis for Equation (2)). Improved sanitation coverage does not necessarily signify the implementation of waste collection and disposal, particularly for some decentralized sanitation systems (e.g. septic tanks and pit latrines). This can be the case in developing nations and less-developed regions within a country with inadequate sanitation infrastructure and management. However, a systems approach to improved sanitation is currently being emphasized to ensure the sustainability of the improvements. In this case, the

provision of access to sanitation services is considered a multi-step process, in which pit latrines or other similar decentralized facilities are parts of the chain and are supported by the collection and transportation of waste as well as its treatment for safe disposal. Further, this analysis also includes a variation to account for the fact that using faecal sludge and urine collected from septic tanks and pit latrines as fertilizers is a common practice in many parts of the world. Consequently, the environmental burdens and benefits of recycling human excreta waste (in terms of sewage sludge) to agriculture are also included in this study. To this end, we assume that all anthropogenic nutrients could enter sanitation and wastewater facilities. Nutrients from human excreta from populations without access to improved sanitation systems were also included and assumed to receive no treatment prior to discharge (Equation (3)). The percentage of the population served by improved sanitation usage was subtracted from 100% to estimate the percentage of the population unserved by improved sanitation facilities. The above statements have been added in the revised text (Pages 28–29, Lines 599–625).

25. Lines 317-322. While Supplementary Table S1 is referenced in the subsequent section, it would be helpful to mention it here as well. Additionally, I would avoid phrasing that states data from more than 60 years (1990-2050) are used. Any projected values (i.e., those going beyond historical datasets) should not be labeled as “data”. I would classify any initial inputs to the overall model as “data”, and I would refer to any projected values as “projections” or “estimates”.

R: We have made these changes throughout the revised manuscript based on the principles you suggested.

26. Lines 335-337. While Supplementary Database S3 is referenced at the end of this section, it would be helpful to mention Table 1 here.

R: We accept this suggestion and have made changes in the revised text (Page 11, Lines 223–226).

27. Line 363. The word “estimate” should be “estimates”.

R: The word ‘estimate’ has been replaced by ‘estimates’ (Page 19, Line 405).

28. Line 366. I am hesitant to call these data “geospatial”. While they do provide geographic diversity from country to country, “geospatial” often refers to data at a finer spatial resolution. I would prefer the term “country-level” or something similar.

R: We accept this suggestion. The word “geospatial” has been replaced with ‘country-level’ (Page 27, Line 583).

29. Line 423. The word “liked” should be “linked”.

R: The typographical error you pointed out has been corrected (Page 30, Line 661).

30. Line 426-428. It is worth noting that, in addition to the uncertainty surrounding the model inputs and results, the planetary boundaries are also associated with substantial zones of uncertainty. In particular, Steffen et al. (2015) report the range for the boundary around global phosphorus discharges to water as being 11-100 Tg P per year.

R: We have added statements to acknowledge the potential uncertainty deriving from the thresholds of the planetary boundaries (Pages 19–20, Lines 418–425).

31. Figure 1. Instead of displaying points with different colors and sizes to represent N and P discharges, the authors might consider including two columns of maps, one for N and one for P, with the countries shaded according to discharge levels. This format might be clearer and make trends easier for the reader to see. In line with my General Comment concerning reframing the results, it might be appropriate to normalize the masses of discharged nutrients relative to population, to account for wide population differences across countries.

R: This comment is constructive, as it allows a better analysis of the findings from the perspective you suggest. However, we would prefer to retain Figure 1 unchanged, as we are sure that it can provide sufficient information for readers in its current form.

32. Figure 2. The authors might consider switching the color scale so that red is larger than yellow. I think that would be more intuitive for readers. Also, again, reframing the results around key drivers of these discharges may lead to a different figure, or a similar one showing slightly different information (e.g., environmental effects relative to population).

R: We have amended the color scale to given the view that red is commonly used to represent large values (see the new version of Figure 2). However, no additional changes have been made to this figure since we have retained the original structure of our analyses in the revised manuscript.

Response to Reviewer 2

1. The manuscript NCOMMS-18-13690-T presents an interesting study on the impact of nutrient (nitrogen and phosphorous) removal from wastewater into different LCA impact categories. The study has a worldwide perspective and considers both spatial and temporal variability. Thus, it allows to identify the change of trends on the global hotspots, i.e. where (geographically) to focus efforts in the incoming decades. The paper is well written, and both the aim and structure are clear. However, if the following issues are considered before publication, they would strengthen significantly the conclusions drawn and the interest and impact of the publication.

R: We highly appreciate your comments and suggestions, which helped us polish our manuscript. We have made careful emendations throughout the manuscript and we present our main point-to-point responses below.

2. Calibration of the model presented is not mentioned in the paper. All results presented, and conclusions withdrawn would be more reliable if some kind of calibration could be made. For example, could any of the modelled estimations of N and P load be compared with real nutrient loads, maybe for some countries or for some regions?

R: We have considered this suggestion carefully. Most projections of anthropogenic nutrient discharge, whether on national, regional, or global scales, mainly relied on assumptions and calculations rather than observations or experiments. Consequently, there were significant disparities in these reported estimates. In this case, using one estimate to calibrate or verify another is unrealistic. To this end, our model to estimate wastewater nutrient discharges was not calibrated against figures reported in the literature. However, we compared the nutrient discharge estimates from this study with those developed in the historical references to test the robustness of our results (Pages 8–9, Lines 161–192).

3. A reference for the IDEAS model should be provided.

R: The IDEAS model was developed by the authors of this study; thus, there are no references to cite in the text.

4. The title of the present paper refers to the impacts of nutrients discharge reduction for 'ecosystem' sustainability but, indeed, ecosystem understood as receiving waters is not considered; the model does not take into account the impacted waters but of course it is not the same to discharge into a receiving media with high or low river flow rates or to discharge into pristine or heavily polluted waters; how could this be

taken into account?

R: We considered this comment seriously and have amended the title of the paper to fit more closely with the objectives and findings of this study. In particular, we changed the term “ecosystem sustainability” to “environmental sustainability”.

5. Only point source pollution due to domestic wastewater is considered in this study, why not industrial wastewater? (although authors recognize its importance for example in page 14, end of the discussion). And industrial wastewater impact can have specially relevance in developing countries where legislation limits are not so strict yet.

R: We agree with you that wastewater generated from industrial sectors could also cause environmental damage; since eutrophic regions are usually collocated with large-scale industrial sites such as food processing plants, pulp and paper plants, refineries, and agriculture. However, including those impacts would require sets of assumptions and approaches to calculations that were completely different from the methodology proposed in this study for domestic wastewater management activities. Moreover, in the last few decades, domestic wastewater has been a key contributor to the increase in nutrients in the aquatic environment. Therefore, we focused our assessment on the broad environmental impacts of reducing nutrient discharge from domestic wastewater and implementing better sanitation and more effective wastewater facilities to achieve particular removal levels.

6. Scenarios: why the 3 initial scenarios (I, II and III) only considers increasing removal efficiencies for N and P but, for example, do not consider an increase on the sanitation coverage (i.e. percentage of people whose wastewater is being treated), even though it is probably a more realistic scenario? Specially in views of the MDGs.

R: We had integrated improved sanitation coverage trends (estimated by the World Health Organization and the United Nations International Children’s Emergency Fund Joint Monitoring Programme for Water Supply and Sanitation) with the three wastewater management regimes (as we now refer to them). However, we have clarified the relevant statements regarding the three regimes in the text (Pages 23–25, Lines 490–544).

7. Indeed the scenarios are not very well described, they are a bit confusing; page 16 mentions 18 scenarios, where can they be seen? their description can be improved. Also, Table 1 is very difficult to understand.

R: We apologize for the confusion you experienced. We have reconstructed the statements regarding the wastewater management regimes (Pages 23–25, Lines 490–

544), sensitivity scenarios (Pages 32–33, Lines 702–726), and other parts of the main text that were as identified being inadequate. In addition, we have also provided additional information in Supplementary Table S7.

8. Scenario 'Stringent nutrient removal' in Table 1 is not Scenario III? Why freshwater use in this scenario experiences a so huge increase (+202%)?

R: In Table 1, the sensitivity analysis scenario “stringent nutrient removal” was not connected to Regime III. Here, we employed the status of the factors such as population, urbanization, food intake, and sanitation coverage in 1990 as a baseline while changing only one factor to its 2050 level, performing the sensitivity analysis of one factor at a time. Thus, the sensitivity analysis scenario “stringent nutrient removal” represented the situation in which nutrient removal efficiencies will be enhanced to reach the levels assumed in Regime III, while other factors are stagnated at the baseline. Implementing this procedure can help us better pinpoint the factors that play a key role in driving the resulting impacts and trends. We have added statements in the revised text (Pages 32–33, Lines 702–726) to clarify the sensitivity analysis means implemented. Regarding why freshwater use experienced so large an increase (+202%) in the sensitivity scenario, this can be ascribed to intensive energy consumption and increased chemical usage owing to the efforts to enhance nutrient removal, since large water footprints are commonly identified throughout the production of energy and chemicals. We appreciate this comment and we have added 1–2 sentences with citations to analyze this finding (Page 14, Lines 297–302).

9. Why only 'conventional hard-tech' treatment solutions (i.e. MLE, MBR. etc. in Table S5) have been considered? Why nature-based solutions (NBS, e.g. constructed wetlands or stabilization ponds), which can be very well fitted in some developing regions, but also in the developed world as a polishing additional treatment step, have not been considered? Operation of this type of NBS provides some advantages with respect to conventional treatment solutions (in terms of energy and chemicals use, maintenance needed, etc.)

R: Improved sanitation coverage would commonly include both centralized and decentralized facilities. In this study, we included several types of engineering solutions for pollution control and further assumed that these approaches were available for both types of wastewater facilities. Centralized wastewater treatment is recognized as a possible solution that allows the sanitation requirements to be satisfied, and many existing biological approaches from centralized wastewater treatment plants could be applied in decentralized systems for smaller towns (and peri-urban areas) and rural communities in developing nations such as China and

India, depending upon the system layout and the operating conditions. Therefore, the values for the life cycle inventories of the engineering-based treatment approaches are assumed to be uniform in terms of modelling the environmental impacts of both treatment systems. Even if natural-based approaches such as constructed wetlands are also seen as potential alternatives, they are excluded in this analysis since ecological approaches are commonly based on local conditions and specific designs, leading to a high level of uncertainty regarding their life cycle impacts across the literature. However, we appreciate the suggestion highly and have added statements to discuss the implications of nature-based solutions in this study (Page 24, Line 523–528).

10. Regarding the future scenarios defined for the sensitivity analysis, why scenarios with renewable energy and/or water reuse and recycling and nitrogen and phosphorus recovery (instead of consuming energy and chemicals for nutrient removal) have not been considered? Nowadays and all over the world, with the circular economy approach and with already with some full-scale implementations, it could be not so unrealistic to consider resource recovery (instead of nutrient removal) for some parts of world, at least partially. Indeed, this is mentioned in the discussion section, pages 12-13, so why not considered as a potential scenario? Also, optimization of 'conventional' treatment technologies will reduce GHG emissions (this has been a research hot topic for many years, so one can also expect operation of existing treatment systems or new treatment systems reducing GHGs emissions).

R: We appreciate the interesting ideas for future studies. The sensitivity scenarios were developed to isolate the influence of each model factor and identify the factors that play a vital role in driving the dynamics of the impacts of wastewater treatment services. The scenarios you mention are highly relevant to emerging solutions that could be potentially implemented to allow a fundamental change in wastewater management from pollution control to a circular economy. We agree with your viewpoint that such transformations do represent future trends that may occur in the water sector, but the impacts of these emerging paradigms are out of the scope of this work. Since a large global population does not have access to adequate wastewater treatment services, our research aimed to investigate the broad environmental impacts of using the available wastewater treatment facilities, particularly for nutrient removal, from a global systems perspective. However, the present study provided scientific support and baseline data for future research efforts investigating the role of water resource recovery paradigms that address both energy and environmental challenges. We have added qualitative discussion of this point to the revised text (Pages 17–18, Lines 361–386).

11. LCI: how CH₄ and N₂O have been estimated? (some published emission factors produce extremely high—so unrealistic—estimated values)

R: In this study, we assumed that CH₄ and N₂O were derived mainly from the handling of wastewater, effluent discharges, and land application of biosolids. Calculation factors for greenhouse gas emissions were obtained from the literature and the probabilistic distributions of the emission factors that took parameter uncertainty into account were used. We have provided extended statements for additional clarity in the text (Page 29, Lines 630–637).

12. LCI: Why anaerobic digestion and biogas uptake are not considered in Table S5?

R: This paper focuses on nutrients; consequently, we only considered nutrient-efficient solutions within the scope of the analysis. Accordingly, biosolids from sanitation and wastewater facilities were assumed to have been disposed of and recycled via land application. In specific, the environmental benefits (avoiding the use of commercial fertilizers) and burdens (emissions of heavy metals to soil) were both considered in the LCA model; relevant life cycle inventory data are given in Supplementary Table S9.

13. LCI: Table S6a includes all construction materials needed for the five treatment technologies considered; however, does the LCA study really includes construction of the treatment plants? As reading the manuscript, it seems only operation is considered (e.g., lines 355-362, page 17, model formulation section); anyway, if considered, why Kg or Tons of used materials are increasing with scenarios moving from I to III?

R: We appreciate the questions you raise. We did include the construction phase of the wastewater treatment approaches in our LCA model analysis. We have revised the text for added clarity (Page 21, Lines 448–450). The alternative wastewater treatment approaches could be applied in all the wastewater treatment regimes proposed in this study; while the capacities of the process configurations would be enhanced with the increases in nutrient removal efficiency, partly in terms of the increased use of raw materials in construction configurations.

14. In many parts of the world eutrophication is largely influenced by diffuse pollution, any idea on how to take into account?

R: In this study, we focused on combining various demographic and socioeconomic factors with nutrient removal regimes to explore the impacts generated by activities to manage nutrients in domestic wastewater. We agree with you that eutrophication could be significantly influenced by diffuse pollution derived from fertilizer use in agriculture. However, studying the resulting environmental impacts would require a

number of assumptions that are completely different from the hypotheses and estimates developed in this study regarding anthropogenic wastewater.

15. Why only 173 individual countries have been considered in this work? which have been the selection criteria? are these countries equally distributed all over the world?

R: In this study, various data sets related to country-level improvements in sanitation coverage, wastewater treatment, life cycle inventory data, food intake, freshwater withdrawal for municipal use, population, and urbanization statistics were collected for 173 countries and territories (collectively referred to as “countries” throughout the analysis). These countries from Africa, the Americas, Asia, Oceania, and Europe were chosen based on the relevant literature and international agency databases, based on the availability of data and principle of consistency across data sets. Overall, the countries selected account for over 70% of the countries and territories in the world and for 99% of the global population; thus, collectively, they reflect the high representativeness of this analysis at a global scale. The country list with analyses of representativeness are provided in Supplementary Table S2. The relevant statements have been added in the revised text (Pages 21–22, Lines 457–473).

16. Fig. 2: regional and temporal variations are very difficult to see; might be better to zoom into a specific region where there is a clear change (e.g. Africa) and another one where there is little change (e.g. Europe)? Page 8, lines 171-174: "...co-benefits could be achieved in most countries from Europe and Caribbean..." how do you see that? how this has been considered?

R: We have responded to these questions you raised, firstly by changing the color scale to ensure that red represents large values. We believe that this would be more intuitive for readers. Figure 2 helped to visualize the overview of the normalized findings regarding the environmental effects of the three nutrient removal regimes at the regional scale during the period 1990–2015. However, we further emphasized the regional variations using Pearson’s correlation indices for the nitrogen cycle and other environmental effects (Supplementary Table S1). Thus, we have extended statements to discuss the findings of the added analyses (Pages 9–10, Lines 195–213).

17. In order to estimate wastewater production, wouldn't it be better to use P.E. instead of inhabitants? which other commercial or industrial activities are considered in the equation 5? or is it really only domestic wastewater?

R: In this study, we used the unit per capita load to estimate wastewater production, which is similar to the metric you mention (population equivalent or PE). We have modified the relevant statements for added clarity (Page 27, Line 584–585).

18. Equation 7 is using $L_{RE,N}$ and $L_{RE,P}$ but text (line 409) refers to $L_{R,N}$ and $L_{R,P}$.

R: We appreciate the careful check from the reviewer. We have made corrections to the text (Page 30, Line 644).

19. Reduction of nutrient loads can cause a reduction in eutrophication but an increase in climate change (CC) and aerosol loadings but, could the authors put these increases in perspective or proportion with respect to other activities causing CC or atmospheric pollution, e.g., future scenarios of transport or mobility or future energy production? Aren't the latter much more relevant to CC than wastewater treatment?

R: Exploring the impacts of public transport and mobility patterns on atmospheric contamination is fairly straightforward. Several past studies have already demonstrated that motor vehicle exhaust emissions create significant unwanted effects on climate change and atmospheric aerosol load. Yet, less emphasis has been placed on the broader environmental responses to wastewater treatment services. This is despite the fact that the use of fossil fuels and emissions of airborne pollutants (such as CO₂, CH₄, and N₂O) throughout the treatment process can increase other seemingly unrelated environmental effects. The core of this work could be a key to solving this question because it indicated the occurrence of problem shifting from aquatic to atmospheric systems. Even if the magnitude of the atmospheric impacts resulting from wastewater treatment services is not as significant as those generated by the transport sector, this study provided valuable information for national and global decision-makers because it identified factors that play an important role in driving these impacts. This could direct future efforts on how to address the interconnected challenges in the water sector, from a systems integration perspective. We appreciate your comments and suggestions. We have added a relevant discussion regarding this point in the revised text (Page 17, Lines 365–369).

20. Oceania's situation is not mentioned in the introduction (page 3).

R: We have made modified the revised text to include this detail (Page 3, Lines 59–60).

Reviewers' comments:

Reviewer #1 (Remarks to the Author):

Although the updated manuscript includes additional text to address some of the points previously raised, some outstanding concerns still exist and should be addressed.

Key outstanding concerns (Items 1-5):

1. Uncertainty analysis: The inclusion of Supplementary Figure S2 and Supplementary Table S12 helps to provide a better sense of the uncertainty analysis. However, I have some remaining concerns:

1A. In Figure S2, error bars for all results (representing the 5th and 95th percentile values from the uncertainty analysis) cover an extremely wide range. They indicate that all life cycle impacts could be zero, or that all impacts could be >100% larger than the global averages. The explanation for this is not adequately clear. Table S12 suggests that life cycle inventory data under each treatment regime were the only parameters varied in the uncertainty analysis, and the assumptions associated with these parameters have (in many cases) almost no uncertainty. For instance, many have min to max ranges that vary by less than 1%. If output ranges are as extreme as they appear to be, the authors should discuss the reasons for this large uncertainty, and whether the uncertainty is too great to draw any practical conclusions from the study.

1B. If life cycle inventory data were the only parameters varied in the uncertainty analysis, then it is critical that other parameters, such as the nitrogen and phosphorus content of protein intake (referred to by the authors as "runoff") and water consumption to withdrawal ratios, also be included. Criteria for including parameters in the uncertainty analysis could be based on Supplementary Table S4. The examples I listed all have average data quality values greater than 2.

1C. The global nutrient discharge values reported in the main text (3.3-4.5 Tg N·yr⁻¹; 0.9-2.0 Tg P·yr⁻¹) are much larger than the global average values shown for nitrogen and phosphorus cycles in Figure S2 (~2x10⁷ kg N; ~1x10⁷ kg P). It is unclear whether these values should be the same, or if they show different things. Also, it is unclear whether the values reported in the figure are annual flows, or if they reflect a different time scale.

2. One-Earth analysis: The discussion surrounding the One-Earth analysis does a better job of putting into perspective the impacts of wastewater treatment across multiple planetary boundary categories. However, the Sankey diagrams in Figure 3 and statements made in the corresponding discussion text still retain the assumption that different impact categories can be added together to estimate a "total" or "aggregated" impact. For example: "The N and P cycles accounted, collectively, for nearly 82% of the total effect in 2010" (lines 336-337; emphasis added). This assumption should not be made. I would suggest the authors replace Figure 3's Sankey diagrams with a different graphic that keeps each boundary category separate. For example, stacked bar charts could be used, with each bar representing a separate impact category and the stack within each bar showing the relative proportion of that impact coming from each continent. In the text, any statements that quantitatively aggregate or compare impacts across multiple categories should be avoided. Instead, trends and relationships can be discussed more qualitatively.

3. Nutrient removal assumptions in Treatment Regime I: The authors have added quite a bit of text to justify their use of globally uniform nutrient removal assumptions. For Treatment Regimes II and III (2010-2030, 2030-2050), these assumptions represent optimistic possibilities for future nutrient removal and may provide a reasonable point of comparison with current conditions. However, the assumptions associated with Treatment Regime I is still questionable. Essentially, this regime can function as a baseline of existing treatment conditions (which can be compared with the future regimes), but applying a uniform nutrient removal rate to all populations around the world does not accurately represent reality. Rather, it creates an initial hypothetical regime that is compared with future hypothetical regimes. Thus, it undermines the potential insight that could be gained from the results, making findings about the actual environmental impacts of increasing nutrient removal (relative to the current or historical reality) questionable. An

alternative, more defensible approach would be use removal rates specifically tailored to individual countries or regions. While it is impossible to be perfectly accurate, this type of approach would provide a baseline that is much more representative of actual conditions.

4. Restructuring of Results and Discussion: In response to my suggestion to restructure the Results and Discussion around key factors that drive nutrient discharges (population, sanitation coverage, protein intake), the authors added a considerable amount of text to the end of the Results and Discussion that examines these drivers. This additional discussion provides useful interpretation of the results. However, the preceding sections of the Results and Discussion still present findings that are likely driven by factors like population and sanitation coverage without mentioning these (relatively obvious) influences. Without integrated explanations of drivers when nutrient discharge and environmental impact trends are reported, it seems as though the most obvious and intuitive explanations for the trends are being ignored throughout the bulk of the discussion. I suggest that the authors include brief explanatory statements in these earlier sections, noting which factors were most important in driving each trend they report.

5. Sensitivity analyses: The authors have clarified their procedures for performing their one-at-a-time sensitivity analysis. However, the process of changing a certain parameter from its 1990 level to its 2050 level while holding everything else constant at 1990 levels remains counterintuitive and unnecessarily simplistic. Additionally, a slightly different procedure is used for parameters that are assumed to remain constant with time. For these parameters, an arbitrary percentage increase or decrease was applied. Mixing these two approaches can create confusion for the reader and makes it more difficult to justify direct comparisons across parameters. As an additional metric of sensitivity, I would suggest that the authors use the outputs from their uncertainty analysis to calculate Spearman's or Pearson's coefficients for all parameters. This approach might provide for more intuitive comparisons across parameters, with results based directly on the variations in parameter values applied in the overall analysis.

Other items that should also be addressed:

6. Generally, the authors are encouraged to proofread the manuscript and Supplementary Information again. There are a number of typographical errors in both.

7. It may be worth noting that nitrogen discharges may not be directly comparable with the planetary boundary for nitrogen (total annual anthropogenic nitrogen fixation). The global phosphorus boundary (P flows to oceans) is more straightforward, but the nitrogen boundary is not directly related to nitrogen discharges into aquatic environments. However, the authors could potentially address this concern by briefly stating that denitrification processes direct dinitrogen gas back to the atmosphere, offsetting some of what humans extract. Recycling nitrogen as fertilizer might also function similarly, in this case reducing nitrogen fixation directly by reducing demand for industrially-produced fertilizer.

8. Lines 54-56. Nitrogen in human metabolism might represent ~15-20% of total anthropogenic production of reactive nitrogen. I'm not sure these two quantities can be said to be "comparable". Perhaps the authors could say that nutrient quantities in wastewater represent a "substantial fraction" of total anthropogenic production.

9. Line 200. "Nutrient-irrelevant effects". I would suggest changing this phrase to "other effects". Nutrients are still relevant, in that their removal drives other environmental impacts in this model.

10. Line 228. "Total food intake" should be changed to "total protein intake".

11. Line 231. Similarly, "vegetable intake" should be changed to "vegetable protein intake" or "plant-based protein intake".

12. Lines 267-279. In this section on dietary intake, the authors could point out that future increases in protein intake are most likely to occur in less developed countries, where population and sanitation coverage will also be increasing. These combined trends could result in large environmental impacts in these countries with less capacity to fund major investments in sanitation infrastructure.

13. Lines 273-274. The factors the authors refer to as nitrogen and phosphorus "runoff" from protein intake are typically used to represent the typical nitrogen and phosphorus content of protein. For example, the baseline nitrogen assumption of 0.13 signifies that protein typically contains 13% nitrogen by mass. This factor does not signify the nitrogen that is "lost" from the body. In adults, intake and excretion of nitrogen and phosphorus are typically at steady state (or nearly so), meaning that any excreted nutrients are replaced by the quantities taken in as food. The authors' calculations are not incorrect. Rather, the terminology being used is somewhat confusing. I would suggest replacing the word "runoff" with "content", and the phrase "interpreted as nutrition loss" should be removed. "Nutrition loss" should also be removed from the section heading. If the authors would like to examine nutrient loss specifically, that should be modeled as a separate factor.

14. Lines 292-296. It would be useful to benchmark these potential changes in emissions from wastewater treatment against total global emissions, to give the reader a better sense of the magnitude of the impact.

15. Lines 318-321. It should be clarified that more stringent regulations on nutrient discharges will protect the aquatic environment, but only from the nutrients present in domestic wastewater. Other (larger) nutrient streams, such as agricultural runoff, will not typically be affected by point-source regulations.

16. Lines 325-327. Again, the authors should clarify that the statement "intensive nutrient removal from wastewater keeps nutrient levels well within the planetary boundaries" is only true with respect to domestic wastewater. In total, humanity has already exceeded the planetary boundaries (or is at least already in the zone of uncertainty) for both nitrogen and phosphorus.

17. Line 327. "The not truly sustainable..." I think one or more words may be missing from this phrase. Also, rather than saying intensive nutrient removal is not truly sustainable, the authors might rephrase this statement to talk about tradeoffs across different dimensions of environmental sustainability (nitrogen cycling, greenhouse gas emissions, etc.).

18. Lines 360-368. In this paragraph, it is important to keep the relative magnitude of impacts in perspective. Saying that we might not want to remove nutrients because the removal process will generate greenhouse gases (or other pollution) is not particularly constructive. It is widely accepted that nutrient removal is important and could have a substantial effect on nutrient cycling. The question is how to achieve greater removal with minimal (or no) increases in other types of pollution.

19. Lines 485-487. "At a certain level, the resulting environmental impacts need to be recalculated, even though some of the sensitivity scenarios presented may be highly unlikely in the future." It is unclear to me what the first part of the sentence is suggesting. Perhaps the phrase "at a certain level" is too vague.

20. Lines 531-532. "...it is not surprising that real-life disparities in regulations restricting nutrient discharge within and across countries." I think one or more words may be missing from this phrase.

21. Lines 649-651. I would suggest adding a statement here (or in another appropriate location) to clarify that food waste was not considered in estimates of protein intake.

22. Lines 653-654. Again, for clarity, I would suggest explicitly stating that this analysis (and other previous work) assumed plant-based protein contains twice as much phosphorus as meat-based protein. This point may not come out clearly from this sentence.

23. Lines 613-616. It is not clear to me how this analysis accounts for potential reuse of collected urine and faecal sludge. Can the authors clarify what is meant by "a variation to account for the fact that faecal sludge and urine collected from septic tanks and pit latrines are commonly used as fertilizer"?

24. Supplementary Table S1. South Africa and Botswana should be listed under southern Africa, not western Africa.

25. Supplementary Table S12. At least two values appear to be incorrect. The mid-range value for sigma R (CP) under "Removal of 95% N and 90% P" and the value for sigma O (CC) under "No treatment" is below the "minimum" value in each of these categories.

Reviewer #2 (Remarks to the Author):

The authors have dealt carefully with most of the points raised by the reviewers (and made corresponding amendments in the manuscript text); however, a couple of points need additional attention: if no reference is given for IDEAS, then additional details about this model are needed; also the fact that anaerobic digestion and biogas production have not been considered is a key point in the sense that it will largely affect GHG emissions and global warming LCA environmental impacts, which is the main aim of the paper; if not considered (due to focus on nutrients), at least some additional statements in the text are needed justifying the absence of this technology but indicating the potential effects on the overall impacts, if considered. Regarding the title, it would probably even be more precise as "wastewater sustainability" instead of "environmental sustainability".

Response to Reviewer 1

1. Uncertainty analysis: The inclusion of Supplementary Figure S2 and Supplementary Table S12 helps to provide a better sense of the uncertainty analysis. However, I have some remaining concerns:

(1) In Figure S2, error bars for all results (representing the 5th and 95th percentile values from the uncertainty analysis) cover an extremely wide range. They indicate that all life cycle impacts could be zero, or that all impacts could be even 100% larger than the global averages. The explanation for this is not adequately clear. Table S12 suggests that life cycle inventory data under each treatment regime were the only parameters varied in the uncertainty analysis, and the assumptions associated with these parameters have (in many cases) almost no uncertainty. For instance, many have minimal to maximal ranges that vary by less than 1%. If output ranges are as extreme as they appear to be, the authors should discuss the reasons for this large uncertainty, and whether the uncertainty is too great to draw any practical conclusions from the study.

(2) If life cycle inventory data were the only parameters varied in the uncertainty analysis, then it is critical that other parameters, such as the nitrogen and phosphorus content of protein intake (referred to by the authors as “runoff”) and water consumption to withdrawal ratios, also be included. Further, criteria for including parameters in the uncertainty analysis could be based on Supplementary Table S4. The examples I listed all have average data quality values greater than 2.

(3) The global nutrient discharge values reported in the main text (3.3-4.5 Tg N·yr⁻¹; 0.9-2.0 Tg P·yr⁻¹) are much larger than the global average values shown for nitrogen and phosphorus cycles in Figure S2 (~2×10⁷ kg N; ~1×10⁷ kg P). It is unclear whether these values should be the same, or if they show different things. Moreover, it is unclear whether the values reported in the figure are annual flows, or if they reflect a different time scale.

R: We appreciate these suggestions and have reperformed the uncertainty analysis (see the new Supplementary Fig. S2). The model parameters with worse data quality (those with an average value of data quality greater than 2, Supplementary Table S4) have been included in the uncertainty accounting, with information related to the probability distributions of the model parameters presented in Supplementary Table S8. Thus, we believe that the concerns raised by the reviewer have been addressed appropriately.

2. One-Earth analysis: The discussion surrounding the One-Earth analysis does a better job of putting into perspective the impacts of wastewater treatment activities

across multiple planetary boundary categories. However, the Sankey diagrams in Figure 3 and statements made in the corresponding discussion text still retain the assumption that different impact categories can be added together to estimate a “total” or “aggregated” impact. For example: “The N and P cycles accounted, collectively, for nearly 82% of the total effect in 2010” (lines 336-337; emphasis added). This assumption should not be made. I would like to suggest the authors to replace Figure 3’s Sankey diagrams with a different graphic that keeps each boundary category separate. For example, stacked bar charts could be used, with each bar representing a separate impact category and the stack within each bar showing the relative proportion of that impact coming from each continent. In the text, any statements that quantitatively aggregate or compare impacts across multiple categories should be avoided. Instead, trends and relationships can be discussed more qualitatively.

R: We accept these suggestions and we have made major revisions to both the figure and the main text. In Fig. 4b, ring diagrams were generated to visualise the contributions of different continents to each effect. In addition, trends and relationships derived from Fig. 4 are discussed more qualitatively in the revised text (page 17–18, lines 366–381).

3. Nutrient removal assumptions in Regime I: The authors have added quite a bit of text to justify their use of globally uniform nutrient removal assumptions. For Regimes II and III (2010-2030, 2030-2050), these assumptions represent optimistic possibilities for future nutrient removal and may provide a reasonable point of comparison with current conditions. However, the assumptions related with Regime I is still questionable. Essentially, this regime can function as a baseline of existing conditions (which can be compared with the future regimes), but applying a uniform nutrient removal rate to all populations around the world does not accurately represent reality. Rather, it creates an initial hypothetical regime that is compared with future hypothetical regimes. Thus, it undermines the potential insight that could be gained from the results, making findings about the actual environmental impacts of increasing nutrient removal (relative to the current or historical reality) questionable. An alternative, more defensible approach would be use of removal rates specifically tailored to individual countries or regions. While it is impossible to be perfectly accurate, this type of approach would provide a baseline that is much more representative of actual conditions.

R: We have reconsidered this comment carefully and we applied an alternative method that is much more representative to simulate actual situations of nutrient removal practice. In particular, we used regional estimates in relation to the prevalence of different wastewater treatment levels (primary and secondary) and

combined them with the nutrient removal rate specifically reported for each treatment level in recent past. Accordingly, we have reconstructed the narratives relative to the three regimes. In brief, the business-as-usual (BAU) regime aligns the prevalence of different wastewater treatment levels (primary and secondary) with their respective estimates at regional level to simulate the actual nutrient removal situations for the period 1990–2010. The secondary-treatment-for-all (STA) regime assumes universal access to secondary wastewater treatment for the period 2011–2030. The universal-tertiary-treatment (UTT) regime represents a more-optimistic likelihood for future nutrient removal and assumes complete usage of tertiary treatment systems across the world for the period 2031–2050. The assumptions made for these new regimes are presented in detail in the text (pages 5, lines 94–105; pages 25–26, lines 542–567). Therefore, we have updated the model results in relation to the analyses of these new wastewater management regimes.

4. Restructuring of Results and Discussion: In response to my suggestion to restructure the Results and Discussion around key factors that drive nutrient discharges (population, sanitation coverage, protein intake), the authors added a considerable amount of text to the end of the Results and Discussion that examines these key drivers. This additional discussion provides useful interpretation of the results. However, the preceding sections of the Results and Discussion still present findings that are likely driven by factors like population and sanitation without mentioning these (relatively obvious) influences. Without integrated explanations of drivers when nutrient discharge and impact trends are reported, it seems as though the most obvious and intuitive explanations for the trends are being ignored throughout the bulk of the discussion. I suggest that the authors include brief explanatory statements in these earlier sections, noting which factors were most important in driving each trend they report.

R: We accepted this suggestion and have reconstructed the relevant statements in the revised text (pages 6–10, lines 122–214; pages 10–12, lines 216–245).

5. Sensitivity analyses: The authors have clarified their procedures for performing their one-at-a-time sensitivity analysis. However, the process of changing a certain parameter from its 1990 level to its 2050 level while holding everything else constant at 1990 levels remains counterintuitive and unnecessarily simplistic. Additionally, a slightly different procedure is used for parameters that are assumed to remain constant with time. For these parameters, an arbitrary percentage increase or decrease was applied. Mixing these two approaches can create confusion for the reader and makes it more difficult to justify direct comparisons across parameters. As an additional metric

of sensitivity, I would like to suggest that the authors use the outputs from their uncertainty analysis to calculate Spearman's or Pearson's coefficients for all model parameters. This approach might provide for more intuitive comparisons across parameters, with results based directly on the variations in parameter values applied in the analysis.

R: We considered this suggestion seriously, together with similar comments raised by the reviewer in the last round of review. Accordingly, we have altered the approach of our sensitivity analyses to make them more intuitive and simplistic. Specifically, we held everything at a given year (1990), changed one of the input parameters by a set percentage (20%), and subsequently compared the model outputs to determine the sensitivity of each model input. We have rewritten the sensitivity analysis procedure for added clarity (pages 34–35, 740–748). Further, we have presented the results of the sensitivity analyses in a more intuitive manner (see the new Fig. 3).

6. Generally, the authors are encouraged to proofread their manuscript (including the Supplementary Information) again. There are a number of typographical errors in both.

R: We have proofread the revised manuscript and the Supplementary Information carefully.

7. It may be worth noting that nitrogen discharges may not be directly comparable with the planetary boundary for N (total annual anthropogenic N fixation). Though the global P boundary (P flows to oceans) is more straightforward, the N boundary is not directly related to nitrogen discharges into aquatic environments. However, the authors could potentially address this concern by briefly stating that denitrification processes direct dinitrogen gas back to the atmosphere, offsetting some of what humans extract. Further, recycling N as fertilizer might also function similarly, in this case reducing N fixation directly by reducing demand for industrially-produced fertilizer.

R: We appreciate this suggestion and we have added the relevant discussion to the text (page 17, lines 359–365).

8. Lines 54-56. Nitrogen in human metabolism might represent roughly 15-20% of total anthropogenic production of reactive nitrogen. I'm not sure these two quantities can be said to be "comparable". Perhaps the authors could say that the nutrient quantities in wastewater represent a "substantial fraction" of total anthropogenic production.

R: We accept the suggestion and have rewritten the relevant statements in the revised

text (page 3, lines 51–55).

9. Line 200. “Nutrient-irrelevant effects”. I would suggest changing this phrase to “other effects”. Because nutrients are still relevant, in that their removal drives other environmental impacts in this model.

R: We accept the suggestion and have replaced the phrase “nutrient-irrelevant effects” with the suggested “other environmental impact factors” (Page 13, Lines 280–281).

10. Line 228. “Total food intake” should be changed to “total protein intake”.

R: We have replaced the phrase “total food intake” with the phrase “dietary protein intake” throughout the manuscript.

11. Line 231. Similarly, “vegetable intake” should be changed to “vegetable protein intake” or “plant-based protein intake”.

R: We have replaced the phrase “vegetable intake” with the phrase “plant-based protein intake” throughout the text.

12. Lines 267-279. In this section on dietary intake, the authors could point out that future increases in protein intake are most likely to occur in less developed countries, where population and sanitation coverage will also be increasing. These combined trends could result in large environmental impacts in these countries with less capacity to fund major investments in sanitation infrastructure.

R: We appreciate the suggestion and have added a related discussion to the text (pages 19–20, lines 437–442).

13. Lines 273-274. The factors the authors refer to as nitrogen and phosphorus “runoff” from protein intake are typically used to represent the typical nitrogen and phosphorus content of protein. For example, the baseline nitrogen assumption of 0.13 signifies that protein typically contains nearly 13% nitrogen by mass. This factor does not signify the nitrogen that is “lost” from the body. In adults, intake and excretion of nitrogen and phosphorus are typically at steady state (or nearly so), meaning that any excreted nutrients are replaced by the quantities taken in as food. The authors’ calculations are not incorrect. Rather, the terminology being used is somewhat confusing. I would like to suggest replacing the word “runoff” with “content”, and the phrase “interpreted as nutrition loss” should be removed. “Nutrition loss” should also be removed from the section heading. If the authors would like to examine nutrient loss specifically, that should be modeled as a separate factor.

R: We accept the suggestion and have replaced the word “runoff” with “content”. In addition, we have improved the statements in the revised text to clarify the model parameters (page 32, lines 687–693).

14. Lines 292-296. It would be quite useful to benchmark these potential changes in emissions from wastewater treatment against total global emissions, to give the reader a better sense of the magnitude of the impact.

R: We appreciate this suggestion, but the discussion pointed out by the reviewer was removed with the revision of the related statements (pages 15–16, lines 323–327).

15. Lines 318-321. It should be clarified that more stringent regulations on nutrient discharges will protect the aquatic environment, but only from the nutrients present in domestic wastewater. Other (larger) nutrient streams, such as agricultural runoff, will not typically be affected by point-source regulations.

R: We accept the suggestion and have amended the relevant statements for clarity (page 17, lines 354–357).

16. Lines 325-327. Again, the authors should clarify that the statement “intensive nutrient removal from wastewater keeps nutrient levels well within the planetary boundaries” is only true with respect to domestic wastewater. In total, humanity has already exceeded the planetary boundaries (or is at least already in the zone of uncertainty) for both nitrogen and phosphorus.

R: We accept the suggestion and have improved the relevant statements for clarity (page 16, lines 346–348).

17. Line 327. “The not truly sustainable...” I think one or more words may be missing from this phrase. Also, rather than saying intensive nutrient removal is not truly sustainable, the authors might rephrase this statement to talk about tradeoffs across different dimensions of environmental sustainability (nitrogen cycling, greenhouse gas emissions, etc.).

R: We accept the suggestion and have improved the relevant statements for clarity (page 17, lines 357–359).

18. Lines 360-368. In this paragraph, it is important to keep the relative magnitude of impacts in perspective. Saying that we might not want to remove nutrients because the removal process will generate greenhouse gases (or other pollution) is not particularly constructive. It is widely accepted that nutrient removal is important and could have a substantial effect on nutrient cycling. The question is how to achieve greater removal

with minimal (or no) increases in other types of pollution.

R: We accept the suggestion and have improved the relevant statements for added clarity (pages 19, lines 399–400; lines 409–412).

19. Lines 485-487. “At a certain level, the resulting environmental impacts need to be recalculated, even though some of the sensitivity scenarios presented may be highly unlikely in the future.” It is unclear to me what the first part of the sentence is suggesting. Perhaps the phrase “at a certain level” is too vague.

R: We accept the suggestion and have removed the relevant statements.

20. Lines 531-532. “...it is not surprising that real-life disparities in regulations restricting nutrient discharge within and across countries.” I think one or more words may be missing from this phrase.

R: We accept the suggestion and have improved the relevant statements for clarity (page 26, lines 556–559).

21. Lines 649-651. I would suggest adding a statement here (or in another appropriate location) to clarify that food waste was not considered in estimates of protein intake.

R: We accept the suggestion and have improved the relevant statement for clarity (page 32, lines 693–694).

22. Lines 653-654. Again, for clarity, I would suggest explicitly stating that this analysis (and other previous work) assumed that plant-based protein contains twice as much phosphorus as animal-based protein. This point may not come out clearly from this sentence.

R: We accept the suggestion and have improved the relevant statement for clarity (page 15, lines 312–313).

23. Lines 613-616. It is not clear to me how this analysis accounts for potential reuse of collected urine and faecal sludge. Can the authors clarify what is meant by “a variation to account for the fact that faecal sludge and urine collected from septic tanks and pit latrines are commonly used as fertilizer”?

R: We accept the suggestion and have rewritten the relevant statements for clarity (page 30, lines 656–657).

24. Supplementary Table S1. South Africa and Botswana should be listed under southern Africa, not western Africa.

R: South Africa and Botswana were indeed listed under southern Africa.

25. Supplementary Table S12. At least two values appear to be incorrect. The mid-range value for sigma R (CP) under “Removal of 95% N and 90% P” and the value for sigma O (CC) under “No treatment” is below the “minimum” value in each of these categories.

R: We appreciate this correction and have double checked the values listed in the Supplementary Information.

Response to Reviewer 2

1. If no reference is given for IDEAS, then additional details about this model are needed.

R: We appreciate this suggestion and the methodology for the IDEAS model (including framework, scope, algorithms, assumptions, and data sources among other common components) is presented in detail in the main text and Supplementary Information.

2. The fact that anaerobic digestion and biogas production have not been considered is a key point in the sense that it will largely affect GHG emissions and global warming LCA environmental impacts, which is the main aim of the paper; if not considered (due to focus on nutrients), at least some added statements in the text are needed justifying the absence of this technology but indicating the potential effects on the overall impacts, if considered.

R: We considered this comment carefully and we have added a relevant discussion in the revised text (page 22, lines 466–470).

3. Regarding the paper title, it would probably even be more precise as "wastewater sustainability" instead of "environmental sustainability".

R: We appreciate this suggestion. However, we contend that the phase "environmental sustainability" is more appropriate for the study, as we attempted to explore potential synergies and tradeoffs across different dimensions of environmental sustainability.

REVIEWERS' COMMENTS:

Reviewer #3 (Remarks to the Author):

The study addresses an important issue of our times: Synergies and trade-offs from nutrient discharge reduction for environmental sustainability. With water stress and environmental pollution becoming two major challenges to the human society as well as the ecosystems, this study on investigating the potential synergies and trade-offs concerning reduction of N and P discharges on the global scale is significant. It sheds lights to the understanding of the interconnections among different earth systems and is helpful for supporting the informed global and national policies and governance in the relevant sectors.

As indicated in the review invitation email from the editor, the manuscript has undergone two rounds of review processes in the journal. I am invited to review the third version of the manuscript. I first went through the reviewers' reports from the second round of review to get a general idea about the comments of the reviewers and check if the authors have adequately responded and addressed the comments. Reviewer 1 raised many comments. However, only the first 6 comments require some major work in the revision. The rest are for the editorial changes and clarifications. The comments from Reviewer 2 are few and mainly for clarification.

By reading through all the responses and the revisions in the manuscript and SI, I am glad to find that the authors have carefully addressed all the comments. Based on my knowledge in the subject field, I think the responses are satisfactory. I do not have further questions regarding the methods, analysis, findings and discussions. Although uncertainties in the results could still be relatively large, they may be considered as 'normal', given the global scale analysis with limited data and very diverse socio-economic and technological conditions across the countries. The authors made much effort in addressing uncertainties and tried to contain them to reasonable ranges. I am satisfied with their treatment to uncertainties.

I have only a few editorial comments to the manuscript.

Abstract, the important driving factors are identified, e.g., population and dietary protein intake. However, the specific synergies and tradeoffs are not highlighted in Abstract in relation to the identified factors. I strongly suggest the authors to report some concrete key findings (with numbers) concerning synergies and tradeoffs to enrich the information in Abstract, and to attract audience to go to read the detail of the paper.

P.6, lines 116-117. The factor 'food and water supplies' is not included in the key drivers shown in P.12, lines 248-250. The authors should replace it with a more relevant variable(s), e.g., food and water (consumption) related factors.

P.13, lines 280-282. 'Consequently, it is estimated that by 2050, nearly 90% of the global population will live in less developed countries'. I understand that this is cited from the UN source. However, it may be worth noting that this percentage apparently did not take into consideration China's very fast development and its aspiration to be a member of the developed world by the middle of this century. Given there is high possibility for China to realize its ambition, 90% may be an arguable number. The authors should put a note here.

Climate change is a very broad field. In this paper, CC refers to GHGs generated from the wastewater treatment processes. It would be more appropriate to directly use GHGs, instead of CC, in my opinion.

P.2, line 37. I think it is better to use 'have' here, instead of 'had'. You are projecting to 2050.

Response to Reviewer 3

1. The study addresses an important issue of our times: Synergies and trade-offs from nutrient discharge reduction for environmental sustainability. With water stress and environmental pollution becoming two major challenges to the human society as well as the ecosystems, this study on investigating the potential synergies and trade-offs concerning reduction of N and P discharges on the global scale is significant. It sheds lights to the understanding of the interconnections among different earth systems and is helpful for supporting the informed global and national policies and governance in the relevant sectors. As indicated in the review invitation email from the editor, the manuscript has undergone two rounds of review processes in the journal. I am invited to review the third version of the manuscript. I first went through the reviewers' reports from the second round of review to get a general idea about the comments of the reviewers and check if the authors have adequately responded and addressed the comments. Reviewer 1 raised many comments. However, only the first 6 comments require some major work in the revision. The rest are for the editorial changes and clarifications. The comments from Reviewer 2 are few and mainly for clarification. By reading through all the responses and the revisions in the manuscript and SI, I am glad to find that the authors have carefully addressed all the comments. Based on my knowledge in the subject field, I think the responses are satisfactory. I do not have further questions regarding the methods, analysis, findings and discussions. Although uncertainties in the results could still be relatively large, they may be considered as "normal", given the global scale analysis with limited data and very diverse socio-economic and technological conditions across the countries. The authors made much effort in addressing uncertainties and tried to contain them to reasonable ranges. I am satisfied with their treatment to uncertainties. I have only a few editorial comments to the manuscript.

R: We appreciate your commendation of the significance and novelty of our present work. In addition, we highly appreciate the editorial comments you raised, as you have indeed helped us improve the manuscript. We have revised the manuscript and provided our point-by-point response below.

2. Abstract, the important driving factors are identified, e.g., population and dietary protein intake. However, the specific synergies and tradeoffs are not highlighted in Abstract in relation to the identified factors. I strongly suggest the authors to report some concrete key findings (with numbers) concerning synergies and tradeoffs to enrich the information in Abstract, and to attract audience to go to read the detail of the paper.

R: We appreciate these suggestions and we have revised the Abstract to increase impact and discoverability of our present work.

3. Page 6, lines 116-117. The factor “food and water supplies” is not included in the key drivers shown in P.12, lines 248-250. The authors should replace it with a more relevant variable(s), e.g., food and water (consumption) related factors.

R: We accept this suggestion and we have revised the relevant term in the text (Page 6, Line 139).

4. Page 13, lines 280-282. “Consequently, it is estimated that by 2050, nearly 90% of the global population will live in less developed countries”. I understand that this is cited from the UN source. However, it may be worth noting that this percentage apparently did not take into consideration China’s very fast development and its aspiration to be a member of the developed world by the middle of this century. Given there is high possibility for China to realize its ambition, 90% may be an arguable number. The authors should put a note here.

R: We consider this suggestion carefully and we have provided a note in the revised text (Page 13, Lines 304–306)

5. Climate change is a very broad field. In this paper, CC refers to GHGs generated from the wastewater treatment processes. It would be more appropriate to directly use GHGs, instead of CC, in my opinion.

R: We appreciate this suggestion and we have provided an appropriate note to specify the definition of climate change for added clarity (Page 6, Lines 134–135).

6. Page 2, line 37. I think it is better to use “have” here, instead of “had”. You are projecting to 2050.

R: We accept this suggestion and we have made relevant modification.